# Exploring the biogeophysical limits of global food production under different climate change scenarios

Philipp de Vrese[1], Tobias Stacke[1], and Stefan Hagemann[2]

[1]Max Planck Institute for Meteorology, The Land in the Earth System, Hamburg, 20146, Germany
[2]Helmholtz-Zentrum Geesthacht, Institute of Coastal Research, Geesthacht, 21502, Germany

*Correspondence to:* Philipp de Vrese (philipp.de-vrese@mpimet.mpg.de)

**Abstract.** An adapted Earth system model is used to investigate the limitations that future climate and water availability impose on the potential expansion and productivity of croplands. The model maximizes the cropland area under prevailing climate conditions and accounts for an optimized, sustainable irrigation practice, thus allowing to consider the two-way feedback between climate and agriculture. For three green house gas concentration scenarios (RCP2.6,RCP4.5,RCP8.5), we show that the total cropland area could be extended substantially throughout the 21st century, especially in South America and sub-Saharan Africa, where the rising water demand resulting from increasing temperatures can largely be met by increasing precipitation and irrigation rates. When accounting for the $CO_2$ fertilization effect, only few agricultural areas have to be abandoned owing to declines in productivity, while increasing temperatures allow to expand croplands even into high northern latitudes. Without the $CO_2$ fertilization effect there is no increase in the overall cropland fraction during the second half of the century but areal losses in increasingly water-stressed regions can be compensated by an expansion in regions that were previously too cold. However, global yields are more sensitive and, without the benefits of $CO_2$ fertilization, they may decrease when green house gas concentrations exceed the RCP4.5 scenario. For certain regions the situation is even more concerning and guaranteeing food security in dry areas in Northern Africa, the Middle East and South Asia will become increasingly difficult, even for the idealized scenarios investigated in this study.

## 1 Food supply and climate change

The question of how many people Earth may accommodate is anything but new, and increasing population sizes have been connected to societal problems by as early as the 18th century (Cohen, 1995a; Van Den Bergh and Rietveld, 2004; UN, 2012). Different factors determine the maximum number of Earth's inhabitants, one of the most elemental being the availability of food. Here, suitable soils, energy and fresh water constitute the most essential factors determining food security, as the largest extent of our food supply originates from agriculture. The availability of these inputs, especially that of fresh water, depends on climatic conditions and understanding their vulnerability to climate change is a major challenge of climate research (Marotzke et al., 2017). At the same time, agricultural activity has substantial impacts on climate through the alteration of land-surface characteristics and the redistribution of water via irrigation.

Numerous studies have investigated the planet's human carrying capacity as a function of the potential food supply (Cohen, 1995a, b; Van Den Bergh and Rietveld, 2004; Franck et al., 2011; UN, 2012; Sakschewski et al., 2014), the effect of climate change on agriculture and water resources (Rosenzweig and Parry, 1994; Jones and Thornton, 2003; Parry et al., 2004; Fischer et al., 2005; Elliott et al., 2013; Haddeland et al., 2013; Konzmann et al., 2013; Wada et al., 2013; Rosenzweig et al., 2014; Pugh et al., 2016) and the climate impacts of deforestation and expanding agriculture (Lobell et al., 2006; Sacks et al., 2009; Cook et al., 2011; Lawrence and Vandecar, 2014; Devaraju et al., 2015; Alkama and Cescatti, 2016; de Vrese et al., 2016; Lawrence et al., 2016; Lejeune et al., 2017). However, the existing studies have an unilateral focus. Climate projections, and even studies that focus on irrigation under changing climate conditions, omit constraints related to water availability and important interactive changes in agricultural areas and practices. In turn, studies that use agricultural or hydrological models to estimate potential yields and future water availability, use prescribed climate conditions that were generated without taking into account the entirety of relevant land-use changes. By investigating the above aspects in isolation, decisive feedbacks are neglected.

An Earth System Model (ESM) offers a basic framework capable of projecting future changes in Earth's climate as well as changes in natural vegetation. To account for the two-way feedback between climate and agriculture, we extended this framework, i.e. for the Max-Planck-Institute for Meteorology's ESM (Raddatz et al., 2007; Jungclaus et al., 2013; Stevens et al., 2013) (MPI-ESM), to include interactive land-cover and land-use changes that maximize global crop yields under prevailing climatic conditions (Fig. 1). In the approach, the spatial extent of cultivated areas is modelled as a function of climatic conditions as well as the agricultural water supply. In regions where conditions allow for at least a minimum productivity, i.e. the crops' net primary productivity (NPP) corresponds to a yield of at least $\approx 250 \, \mathrm{t \, km^{-2} (canopy) \, year^{-1}}$, the cultivated area is extended incrementally until all cultivable areas are occupied, i.e. the land not limited by soil or terrain constraints. In regions in which the NPP falls below this threshold, the area under crops declines. The NPP was also used to estimate the potential food production, by assuming that the changes in crop yields are proportional to changes in the plants' NPP. To estimate the potential food production on a hydrologically sustainable basis, future water withdrawals are limited to the fraction of renewable fresh water which exceeds environmental requirements. Here, it is assumed that about a third of the long-term mean flow is required to ensure ecological stability and may not be withdrawn (Pastor et al., 2014). Water for irrigation is removed from the river network and stored in a dedicated reservoir. When required, the water is applied to the soil, from where it evaporates, is taken up by plants and transpired or returned to the river via subsurface runoff (for more details on the methodology see Sec. 4). Together with the changes in the surface-atmosphere exchange of energy and moisture that result from alterations of the surface characteristics, this closes the feedback loop between land-use and climate (Fig. 2).

We used this adapted model to investigate the climate-agriculture dynamics during the 21st century that result from the maximization of the cropland area under different atmospheric green house gas (GHG) concentration scenarios (Fig. 3b, Tab. 1 and Sec. 4). The simulations cover the period 1995 - 2114 and were forced according to three representative concentration pathways (RCP, Meinshausen et al. 2011; van Vuuren et al. 2011) that assume a peak and a subsequent decline of emissions

until 2020 (RCP2.6) and 2040 (RCP4.5) as well as an ongoing increase in emissions (RCP8.5). They use a temporal resolution of 450 seconds, a horizontal resolution of T63 (1.9° × 1.9°) and vertical resolution of 47 atmospheric model levels.

The focus of this investigation is on the global crop yields that are achievable under future climate conditions and, in the following analysis, we will show the potential expansion of cultivated areas, the changes in global yields and how these relate to future food security. The effects of changes in irrigated and rainfed cropland area on climate will only be discussed very briefly as their detailed analysis goes beyond the scope of this study. The present framework targets biogeophysical feedbacks, with a special focus on the hydrological cycle, while other important social, political, economic, ecological and technological

considerations are being neglected. Thus, it is important to note, that the below results merely pertain to the development of cropland areas and yields in a highly idealized scenario but not necessarily to real-world potentials, the latter of which may be much more limited by factors such as economic costs, fertilizer availability, the need to minimize environmental degradation, habitat and biodiversity loss, the limited capabilities of existing irrigation systems, dietary shifts and the competition between food and energy demands. An estimate of actual agricultural potentials requires a more comprehensive framework that is

capable of representing all these factors, e.g. by integrating a broader range of models including dedicated crop, economic and ecosystem models.

## 2   Results

### 2.1   Cropland area

In 2005, croplands covered an area of roughly $14 \cdot 10^6 \mathrm{km}^2$ (see supplementary material Fig. S1) (Hurtt et al., 2011). This corre-

sponds to about 9 % of the global land surface and to 23 % of the potentially cultivable land (Fig. S1)(IIASA and FAO, 2012). Our results show that, for some arid regions, especially in Northern Africa, the Middle East as well as South and Central Asia, it may be impossible to maintain the current cropland area (see below). However, on the global scale, these potential losses can be more than compensated by an expansion of croplands, especially in Sub-Saharan Africa, South America and in higher northern latitudes. In the simulations, the cropland area can be tripled to roughly $38 - 42 \cdot 10^6 \mathrm{km}^2$, and almost three quarters of all cultivable land can be farmed by the beginning of the next century (Fig. 3a, Tab. 1). This expansion would require converting areas into croplands regardless of their present function in the Earth system, including those whose cultivation is highly debatable with respect to biodiversity and terrestrial carbon stores, e.g. the Amazon rainforest. There is no consensus on the extent of protected areas required to maintain the planet's ecological stability and consequently also no consensus on the area that, from an ecosystem perspective, could be transformed into croplands. Merely omitting areas presently placed under

5     protection (Juffe-Bignoli et al., 2014; UN, 2016) and those covered by tropical forests from the analysis already reduces the cultivated area in 2100 by roughly 15% (Sec. 4 and Fig. S2). However, it is most likely that ensuring Earth's habitability will require the protection of a much larger fraction of the land surface (Sec. 3).

**Table 1. Experimental overview** The table provides an overview over the simulations performed in the context of the present study. The upper part shows the most important characteristics of the simulations' setup, specifying the crop management scheme, i.e. the expansion and decline of cultivated areas and the irrigation scheme, the representative concentration pathways (RCP) to which the green house gas (GHG) forcing corresponds, and which $CO_2$ concentration was used in the model's vegetation scheme. The lower part summarizes the most important results, namely the simulated global extent of cultivated areas and the corresponding supportable population size ($K_{hum}$) in the year 2100.

Setup

| Sim. | Cropland area | Irrigation | GHG | $CO_2$ Veg. |
|---|---|---|---|---|
| REF | Prescribed based on present-day cropland | Non-sustainable | - | - |
| RF45 | Dynamic; based on climatic conditions | No irrigation | RCP4.5 | RCP4.5 |
| RF45* | Same as RF45 | Same as RF45 | RCP4.5 | 380.0 ppmv |
| IR26 | Dynamic; based on climatic conditions and the availability of water for irrigation | Sustainable | RCP2.6 | RCP2.6 |
| IR26* | Same as IR26 | Same as IR26 | RCP2.6 | 380.0 ppmv |
| IR45 | Same as IR26 | Same as IR26 | RCP4.5 | RCP4.5 |
| IR45* | Same as IR45 | Same as IR45 | RCP4.5 | 380.0 ppmv |
| IR85 | Same as IR26 | Same as IR26 | RCP8.5 | RCP4.5 |
| IR85* | Same as IR85 | Same as IR85 | RCP8.5 | 380.0 ppmv |

Simulated crop area and supportable population size

| Sim. | Cultivated area in 2100 | $K_{hum}$ in 2100 |
|---|---|---|
| REF | $14^{\#} \cdot 10^6 km^2$ | $6.5bn^{\#}$ |
| RF45 | $39 (33^+) \cdot 10^6 km^2$ | $19bn (16bn^+)$ |
| RF45* | $36 (31^+) \cdot 10^6 km^2$ | $15bn (12bn^+)$ |
| IR26 | $38 (33^+) \cdot 10^6 km^2$ | $20bn (16bn^+)$ |
| IR26* | $38 (33^+) \cdot 10^6 km^2$ | $17bn (14bn^+)$ |
| IR45 | $40 (34^+) \cdot 10^6 km^2$ | $22bn (17bn^+)$ |
| IR45* | $38 (33^+) \cdot 10^6 km^2$ | $17bn (14bn^+)$ |
| IR85 | $42 (36^+) \cdot 10^6 km^2$ | $27bn (22bn^+)$ |
| IR85* | $38 (33^+) \cdot 10^6 km^2$ | $15bn (12bn^+)$ |

[#] Note that for the reference simulations the values correspond to the year 2005 and the actual population size is provided.

[+] The values in brackets indicate the cultivated area and $K_{hum}$ when omitting protected areas from the analysis, i.e. areas presently placed under protection (Juffe-Bignoli et al., 2014; UN, 2016) and those covered by tropical forests (as of 2005).

The extent of the present-day cropland is not in equilibrium with the constraints resulting from today's climate. Wide areas could be cultivated without requiring any changes in the conditions, i.e. temperatures and precipitation rates are already in a

favourable range at the beginning of the century, and the largest potential for expansion is given in latitudinal zones in which crops are already being grown (Fig. 3a; right panel). At the same time, irrigation in dry regions relies partly on non-renewable sources and the irrigatable area is reduced substantially when the water withdrawals are limited to the fraction of renewable freshwater that exceeds environmental requirements (Fig. S4). The reduction in irrigation also affects the adjacent rainfed agriculture as the resulting decline in evapotranspiration leads to a reduced moisture recycling, lowering precipitation rates. During roughly the first 25 years of the simulation, the net-change in cropland area is less determined by changes in the climate and mostly an adjustment to the limits given by the simulations' initial conditions. Here, the area that can be cultivated, without requiring any change in the conditions is far larger than the area that has to be abandoned as irrigation is limited to sustainable amounts. Consequently, there is a strong increase in cropland area to about $35 \cdot 10^6 \text{km}^2$ ($30 \cdot 10^6 \text{km}^2$ when excluding protected areas). After 2025, the global cropland area increases at a much slower rate as their expansion and decline is now dependant upon changes in GHG concentrations and climatic conditions.

All simulations exhibit a temperature increase, ranging between roughly 1 K for RCP2.6 (IR26) and 5.5 K for RCP8.5 (IR85) (Fig. 3c). There are two opposing effects related to rising temperatures. On one hand they can lead to an increased productivity in energy-limited regions, prolonging growing seasons in the (higher) mid and high latitudes, which is especially important for the northern hemisphere, where farming becomes possible even north of 60°N. On the other hand, increasing temperatures raise the crops' water requirements leading to an increased water stress and reduced productivity in water limited regions. Hence, the overall effect of increasing temperatures is also determined by the change in water availability (Fig. 4), thus by the change in precipitation.

The scenarios that exhibit a strong temperature rise also show a substantial increase in precipitation over land (Fig. 3d). For RCP4.5 (IR45) mean precipitation rates increase by up to 20 mm year$^{-1}$ and in IR85 they increase by about 60 mm year$^{-1}$, which amounts to more than 8% of the terrestrial precipitation (as of 2005). Increased precipitation rates do not only reduce the water stress for rainfed crops, but between 2025 and 2100 they also increase the water available for irrigation; globally by roughly 500 km$^3$year$^{-1}$, for IR45, and by almost 2000 km$^3$year$^{-1}$ for IR85 (Fig. 3e). As a consequence, the increased water demand of irrigated and rainfed crops resulting from higher temperatures can be met to the extent that, after 2025, there are only very few areas in the world in which farming becomes unsustainable. This however is only the case when fully accounting for the potential benefits due to the $CO_2$ fertilization effect (CFE; see below). For the simulations with only a small increase in GHG concentrations (IR26) there is no permanent increase in precipitation, i.e. after a peak in the 2040s the rates decline to their initial levels, while the average temperature at the land surface increases by $\approx$1K. Here, the plant's increasing water requirements can not be met everywhere and in some dry regions in South and Central Asia, the Sahel zone and Australia farming becomes unsustainable after 2025 and cropland areas have to be abandoned.

The results show that future climate is substantially impacted by the maximization of irrigation within sustainable limits. The precipitation rates in the RCP4.5 simulations with (IR45) and without (RF45) irrigation differ on average by roughly 15 -

30 mm year$^{-1}$, and in RF45 precipitation is persistently below present-day rates. This is a strong indication that a substantial part of the precipitation rise in the irrigation simulations is attributable to irrigation-precipitation feedbacks (Fig. 3d). Furthermore, for the RCP4.5 scenario, irrigation reduces the simulated 21st-century temperature increase by almost 20% ($\approx 0.5$ K averaged over the global land surface; Fig. 3c), and in irrigated regions the effect can amount to several K. Here, the irrigation induced surface cooling can also have negative consequences, as it may affect regional circulations and reduce the poleward heat transport, shortening the growing season in high northern latitudes (Fig. S5 - S7).

As an aggregate effect of changing conditions, the area in which farming becomes possible after 2025, i.e. mainly due to increasing temperatures in high latitudes, exceeds the cultivated area which has to be abandoned due increasing water stress in dry regions. Consequently, the global cropland area increases by an additional 4 - 7 $\cdot 10^6$km$^2$, with the expansion for RCP8.5 being almost twice as large as for RCP2.6. Here, the simulated expansion of croplands is not only depending on the changes in climate, discussed above, but also on the CFE, i.e. the effect that plants increase their rate of photosynthesis with increasing $CO_2$ levels. This effect is especially relevant for the cultivation of areas in the high northern latitudes, where it allows farming even in regions where growing seasons are still short. Additionally, the CFE effectively reduces the plants' water requirements, as it shortens the time that they are required to open their stomata for the uptake of carbon. As this may severely reduce the crops water requirements, it also affects the sustainability of croplands in arid and semi arid regions. Studies have identified a greening in recent years that is in large parts related to the CFE (Zhu et al., 2016), however, it is very uncertain which role it may play for future crop yields, and it is even possible that it's benefits will be balanced completely by other factors such as nitrogen limitations (Rosenzweig et al., 2014; Smith et al., 2015; Obermeier et al., 2016).

In simulations with the MPI-ESM, the CFE is very pronounced and, to investigate the limits of food production without this highly uncertain effect, we performed an additional set of simulations (IR26*,IR45*, RF45* and IR85*) that are identical to IR26,IR45,RF45 and IR85 but with the plant available $CO_2$ limited to the level of the year 2005, i.e. 380.00 ppmv. Prescribing the $CO_2$ concentration in this manner affects croplands especially for the RCP8.5 scenario. After 2050, the decrease in cultivated area in increasingly dry regions in mid and low latitudes almost completely balances the expansion in high latitudes and there is almost no increase in the global cropland area (Fig. S2). For the other scenarios, omitting the CFE has a similar effect, although much weaker. As a result, the total cropland area is very similar for the three scenarios, and they differ only in the spatial distribution of croplands. In simulations with higher $CO_2$ concentrations the cultivated area in high northern latitudes is larger, while for the low GHG scenarios the area in arid mid and low latitudes is predominantly larger (Fig. S3).

## 2.2 Potential food supply

Changes in climate and raising $CO_2$ levels do not only affect the potential extent of croplands, but also the productivity in existing cultivated areas. Thus, the global aggregate productivity of croplands is much more sensitive to increasing GHG concentrations and the changes in potential yield are not proportional to the increase in cultivated area. When fully accounting for the CFE and utilizing all land resources, production can be increased by a factor ranging between 3.5 and 5. When excluding

protected areas, this factor still ranges between 2.8 and 4 (Fig. 5a,b). The largest yields are simulated for the high concentration scenario and by 2100, yields are more than 40 % larger for RCP8.5 than for RCP2.6, while the cultivated area is only about 8 % larger. Furthermore, global crop production depends strongly on irrigation. Global yields are 16 % larger in the simulations with than without irrigation while the cultivated area is only about 3 % larger.

To estimate how the changes in productivity relate to future food security, the simulated crop yields can be used to obtain a rough estimate of the supportable population sizes ($K_{hum}$; Sec. 4). When assuming the full benefit of the CFE, the largest $K_{hum}$ is given for RCP8.5. In the respective simulations, the global food supply can roughly be quadrupled during the 21st century, which places the respective $K_{hum}$ at about 27bn people (22bn when excluding protected areas). The lower GHG concentrations in the RCP2.6 simulation decrease $K_{hum}$ by almost 8bn, as compared to RCP8.5, a number larger than the planet's current pop-

ulation. Here, the results seem to suggest that the high concentration trajectory is favourable with respect to food production, however, this is only the case if the CFE is as efficient as simulated by the MPI-ESM. When assuming no benefits due to increasing levels of $CO_2$, the maximum crop yield is much smaller for all scenarios and the global food production can only be increased by a factor of about 2.5 (Fig. 5c,d). Furthermore, the maximum crop yield is reached when the extent of croplands has adjusted to present-day conditions and none of the simulations shows an increase after 2030. The effect from omitting

the CFE is strongest for the high concentration scenario and the respective $K_{hum}$ is reduced by about 12bn (as compared to the RCP8.5 simulation including the CFE). Most importantly, after 2030, the RCP8.5 scenario exhibits a decline in yields and towards the end of the simulation it produces the smallest $K_{hum}$ of any of the simulations that account for irrigation. Here, the simulations provide strong evidence that, even with an optimized irrigation practice and the resulting irrigation-precipitation feedbacks, future increases in precipitation do not meet the additional water requirements due to increasing temperatures. Thus,

the simulated changes in climate may not lead to a loss in the total cropland area but they are detrimental to global yields. Given the high level of uncertainty connected to the CFE, the range of climatic conditions that are favourable for food production is likely limited to the conditions resulting from the RCP4.5 scenario.

     The RCPs are consistent with distinct socio-economic pathways that differ strongly with respect to future energy demand and the mix of energy carriers. Included are assumptions about resource availability and climate policies, which determine the contribution of fossil fuels to the energy mix, as well as assumptions about the population development, which strongly affects future energy demands. Additionally, the scenarios take into account different land-cover and land-use change projections

which reflect future food and energy demands as well as policies with respect to reforestation (Meinshausen et al., 2011; van Vuuren et al., 2011). RCP2.6 and RCP4.5 present intermediate scenarios with ambitious emission reductions, which in case of RCP2.6 even include a decline in the use of oil. For RCP2.6 and RCP4.5 the population development corresponds to UN projections assuming a low to medium fertility and life expectancy in the future (UN, 2004, 2015a, b). In contrast, RCP8.5 presents a highly energy-intensive scenario without the implementation of any climate policies. The high energy

demand in this scenario partly results from a strong population growth, which corresponds to a medium to high population trajectory in the UN development scenarios. In order to estimate the level of food security for a given combination of RCP and

population development scenario, the simulated $K_{hum}$ can be compared to the population levels proposed by the UN scenarios. Here, the simulations indicate that the ability to sustain future populations depends heavily on the strength of the CFE. When assuming the full benefits, only the population of the high-fertility (and life-expectancy) scenario may become unsustainable, i.e the respective population trend surpasses $K_{hum}$ as simulated for RCP2.6 and RCP4.5, and that only if protected areas are maintained (Fig. 5a,b). However, without the CFE the food requirements resulting from the high-fertility scenario can not be met by any simulated supply, even if protected areas are converted into croplands (Fig. 5c,d). Also the population of the medium-fertility scenario is very close to $K_{hum}$, indicating that we may need to cultivate almost all non-protected areas and to have a near-perfect system for irrigation in order to meet the future food requirements of this scenario. Here, our findings contradict the RCP's underlying scenarios. These assume that the population increase in RCP8.5 could be sustained without increasing the cropland area beyond $20 \cdot 10^6 km^2$, while the population increase in RCP4.5 could even be met with a substantial decline in the cultivated area (Meinshausen et al., 2011; van Vuuren et al., 2011).

## 3  Discussion

A value for $K_{hum}$ that is based on globally aggregated estimates conceals further problems that occur at smaller scales, as local declines in the food production are masked by increases in other regions. While the cultivated area and crop yields can be increased substantially in most regions of the world, even without the CFE, this is not the case for Northern Africa, the Middle East, South and Central Asia (Fig. 6a,e,i). Here, a large fraction of present-day irrigation originates from non-renewable sources and reducing it to a hydrologically sustainable level causes a decline in the cropland area or a shift from irrigation-based to less productive rainfed agriculture (Fig. 4, S5). In some cases, the consequent yield declines can potentially be balanced by the CFE in combination with changes in local climate. For example, a reduction of irrigation in India increases surface temperatures and the land-sea thermal contrast, strengthening the Indian monsoon. The resulting increase in precipitation helps to mitigate the effects due to the decline in irrigation and for the RCP8.5 scenario the water-availability even increases. Hence, in combination with the CFE, yields in India decline only slightly for RCP2.6 (Fig. 6a,b) and for RCP8.5 they even increase (Fig. 6e,f). However, in many countries in this region the agricultural water supply declines to an extent that can not be compensated and food production decreases distinctly in all scenarios.

The severity of this problem becomes clearer when comparing the simulated changes in food supply to the prognosed population trends (Fig. 6c,g). For RCP2.6 almost none of the countries in the region is able to support the population development of the low-fertility scenario (Fig. 6d) and for the combination of RCP8.5 and high-fertility scenario, the prospect is even more concerning (Fig. 6h). Consequently, sustaining the prognosed population sizes will likely require a vast increase in food imports in the entire region. The situation is even worse without the possible benefits of the CFE. For the RCP8.5 scenario, simulated yields decrease distinctly throughout the region (Fig. 6i), and there are only few countries in which the potential food production doesn't drop below present day levels (Fig. 6j). Thus, the future food supply in many densely populated countries such as India, Pakistan and Bangladesh may not even suffice for the population trend of the low-fertility scenario (Fig. 6k,l). It

is possible that the present study underestimates the potential food production especially as possible technical solutions, such as better adapted crops or large scale desalination efforts, are not being accounted for. On the other hand, the study neglects important constraints, e.g. resulting from fertilizer availability or the limited water-use efficiency of irrigation systems. As, in reality, these will strongly affect future crop yields, the present idealized scenario may likely provide an overly optimistic outlook. This is especially the case for the simulations that assume a large increase in GHG concentrations, i.e. RCP4.5 and RCP8.5, and account for the full benefits of the CFE.

In the study, the crop's general response to changes in climate agrees well with estimates of other studies (Lobell et al., 2011; Asseng et al., 2014; Challinor et al., 2014). When omitting the CFE and effects of irrigation (RF45*), regions that are presently dominated by rainfed agriculture exhibit an average decline in yield per area of about 5 % per K temperature increase, i.e. in grid boxes where 5 % of the area or more were covered by crops in the year 2005 and less than a third of this cropland area was irrigated, a temperature rise of about 2.6 K caused an average reduction in crop yields per area of about 12 %. The yield response to changes in temperature is strongly affected by the study's management assumptions and, in the same regions, the average yield per area increases by about 2 % per K temperature increase, when irrigation is maximized within sustainable limits (IR45*), i.e. for the temperature rise of about 2.1 K we estimated an average increase in crop yields per area of about 5 %. Hence, the assumptions made with respect to future irrigation, including the representation of the resulting climate feedbacks, are one of the reasons why the development of global crop yields under the RCP scenarios is much more positive than in many other studies (Guoju et al., 2005).

Another reason for the high simulated yields, is the model's comparativly strong CFE. Many studies have investigated the effect of increasing atmospheric $CO_2$ concentrations on vegetation (Tubiello et al., 2007). These indicate that there is a substantial photosynthetic response to increasing $CO_2$ levels, i.e. under optimal conditions, doubling the present day $CO_2$ concentrations leads to an increase in photosynthesis of 30 % - 50 % for C3 and 10 % - 25 % for C4 plants. With respect to crop yields, the existing studies exhibit large uncertainties and strong variations between crop types and regions. For $CO_2$ increases similar to the ones assumed by the RCP4.5 scenario, the estimates range from a 2.5 % to a 25 % yield increase per 100 ppmv increase in $CO_2$ (Amthor, 2001; Tubiello et al., 2007; Ainsworth et al., 2008; Asseng et al., 2013; McGrath and Lobell, 2013). In the RCP8.5 scenario, the atmospheric $CO_2$ concentrations towards the end of the century exceed 1000 ppmv. At these levels, the benefits due to additional $CO_2$ are much smaller as even C3 crops are close to (or have already reached) their saturation level. For the rise in $CO_2$ concentrations assumed by this scenario the average yield increase is expected to be below 6 % - 8 % per 100 ppmv increase in $CO_2$ (Parry, 1990; Amthor, 2001; Ainsworth and Rogers, 2007; Ainsworth and McGrath, 2010). In comparison to these studies, which predominantly consider yield increases under optimal conditions, the MPI-ESM simulates a very strong CFE (approximated by the productivity difference between the simulations with and those without increasing the plant-available $CO_2$). In regions that are being farmed at present (grid boxes in which 5 % of the area or more were covered by crops in the year 2005), the simulations for the RCP4.5 scenario that account for irrigation exhibit an average increase in yield per area of about 18% per 100 ppmv increase in $CO_2$. Owing to the higher temperatures and lower water availability, the sim-

ulated strength of the CFE is slightly lower, i.e. about 14% per 100 ppmv increase in $CO_2$, when irrigation is not represented. For the RCP8.5 scenario, our simulations showed an increase of about 10 % per 100 ppmv increase in $CO_2$. These values place the CFE simulated with the MPI-ESM at the higher end of the range of current estimates, in case of the RCP8.5 scenario even
exceeding them, indicating that the model overestimates the strength of the CFE and the resulting crop yields.

In addition to climate effects, weed and insect pests as well as increasing nutrient requirements are expected to reduce the strength of the CFE (Tubiello et al., 2007). Here, constraints due to fertilizer availability present one of the key limiting factors. For example, Rosenzweig et al. (2014) investigated the crop yield response for the RCP8.5 scenario as simulated with
different global gridded crop models. The study showed that yields for the major crop types predominantly increase if no explicit nitrogen limitation was accounted for. However, when nitrogen limitations are introduced and fertilizer application is restricted to present day rates, the effect of CO2 fertilisation is greatly reduced and all major types exhibit a decline in crop yields throughout the low and parts of the mid latitudes (Rosenzweig et al., 2014). In principle di-nitrogen gas provides an unlimited source of nitrogen. However, nitrogen fixation, i.e. the process by which atmospheric nitrogen is made available for
plants, requires high energy inputs. At present, the share of fertilizer production in the global energy consumption is estimated to be around 1% (Vance, 2001; Dawson and Hilton, 2011), and the fertilizer requirements as proposed in this study could easily increase this share to more than 5%. In case of phosphorus the situation is more difficult as it is, effectively, a non-renewable resource and our supply stems from mines which are located in only a few countries. The size of the phosphate rock deposits is highly uncertain and by far the largest deposits have only since recent been included when taking stock. Given our current use
of phosphorus, these known resources would last for the next 400 - 800 years (Cordell et al., 2009; Dawson and Hilton, 2011). With the increase in fertilizer demand, as proposed by this study, the deposits of phosphate rock may not last long beyond the investigated period. Industrial agriculture, even on the present-day scale, is not possible without phosphorus fertilization and productivity would quickly diminish to the level prior to the agricultural revolutions of the 19th and 20th century if our resources are exhausted. Hence, the future food supply will strongly depend on how much energy is available for the production of fertilizers and how effectively nutrients can be recycled.

But even if sufficient fertilizers could be provided, this would increase other problems related to their application. The present-day fertilizer use already has strong detrimental impacts on the ecosystems in certain regions, where an excess of the
respective elements can leave entire lakes, rivers and coastal stretches uninhabitable to plants and animals (Vitousek et al., 1997; Smith, 2003; Rockström et al., 2009). As a consequence, it has been suggested that the extent of croplands should not surpass 15% of the global ice-free land surface (Rockström et al., 2009; Steffen et al., 2015). Any further expansion could bring the planet to a tipping point, e.g due to hypertrophication resulting from increased use of fertilizers and the loss of biodiversity. This would mean that we have to retain agricultural expansion far below the limits set by climatic conditions.
Limiting croplands to 15% of the global ice-free land surface, would roughly halve the potential cropland area as estimated by this study, i.e. about a third of the ice-free land surface, resulting in similar decreases in crop yields and food security. Additionally, the study's assumption that per capita food requirements will remain at present day levels may also contribute to

an overestimation of the level of food security. Dietary shifts are expected to double global food requirements by 2050 while the population is only expected to increase to about 9bn (Godfray et al., 2010). It is highly doubtful whether this dietary shift and population increase could be sustained without expanding the cultivated areas beyond the safe limit of 15%. Here, our results indicate that only a very strong CFE could lead to the necessary increase in crop yields. Additionally, it would require shifting cultivated areas to the most productive regions, mostly in sub-Saharan Africa, South America and South East Asia, and to provide an almost perfect irrigation system (Fig. 7).

## 4 Methods

For this study, the MPI-ESM was equipped with a newly developed dynamical crop-management scheme and a new water-management scheme. The former consists of a cultivation scheme, in which the extent of cultivated areas is determined, a routine for harvesting and a scheme for simulating irrigation, while the latter contains a routine to determine the environmental flow requirements, a routine to dynamically determine the size of reservoirs and a routine to simulate water withdrawals and releases. A detailed description of these schemes is provided below (see also Fig. 1), followed by a description of the simulations and the analysis that where performed in the context of this study.

### 4.1 Model: Crop Scheme

#### 4.1.1 Cultivation

The cultivation scheme determines the fractional cover of crops, based on their productivity, the temporal distributions of surface temperatures and the availability of water. As an upper limit of annual land-use change, the combined increase in irrigated and non-irrigated crops may not exceed 2.2 % of the grid-box area (GBA). This value was derived from historical expansion rates provided by the Land-Use Harmonization project (Hurtt et al., 2011). To obtain a large, yet plausible value, we chose the 99th percentile (spatial and temporal) from the period 1500 - 2000 of globally available historical expansion rates. Whenever the potential increase in cultivated area surpasses this maximum rate, the cultivated area is divided between irrigated and rainfed crops according to the plants' net primary productivity (NPP). This is done in order to facilitate the expansion of the more productive technique (rainfed or irrigated) and also to reduce the amount of irrigation whenever it does not lead to a clear advantage in terms of NPP. Additionally, the decision of planting C3 or C4 crops is also based on the previous year's NPP of the respective types.

Irrigated crops can exist in all grid-boxes with growing seasons of at least 1300 growing degree days ($^{\circ}$C days). The grid-box fraction potentially covered by irrigated crops is determined based on the volume of the grid-box's water reservoir (see below) and the irrigation requirements per unit irrigated area. The potentially irrigated area is calculated such that the reservoir's content should allow irrigation for a three-month period (assuming the irrigation demands of previous years). In grid-boxes without previous irrigation, the water requirements are set to $1\,\mathrm{m}^3\,\mathrm{m}^{-2}$. To buffer strong variations in the water supply, the size

of the reservoirs is determined based on the multi-year mean streamflow (see below). Additionally, it was assumed that areas that were previously irrigated, are turned into rainfed areas whenever there is a strong decline in the available water. Hence,

cultivated areas are only abandoned when the conditions do not permit to grow rainfed crops.

The extent of rainfed crops is estimated based on their productivity (In JSBACH, i.e. the land-surface scheme of the MPI-ESM, the NPP is calculated for each PFT in every non-glacier land grid-box). The dynamical cultivation scheme increases or decreases the GBA potentially covered by rainfed crops by a rate that depends on the number of previous years in which a

certain NPP threshold is exceeded. This rate is increased by $1.1\,\%(\text{GBA})\text{year}^{-1}$ for every year that the NPP of rainfed crops exceeds the threshold, until the maximum expansion rate of $2.2\,\%(\text{GBA})\text{year}^{-1}$ is reached. For grid-boxes in which the NPP falls short of the threshold, the expansion/decline rate is decreased by $1.1\,\%(\text{GBA})\text{year}^{-1}$, until the maximum decline rate of $-2.2\,\%(\text{GBA})\text{year}^{-1}$ is reached. The NPP threshold was chosen to correspond to a crop yield of $250\,\text{t}\,\text{km}^{-2}(\text{canopy})\,\text{year}^{-1}$ (see below).

Whenever there is an increase in the cultivated area, the cover fractions of all other tiles, i.e. different subareas within a grid-box that are assumed to have homogeneous characteristics, are reduced in a hierarchical order. The cover fraction of natural vegetation, i.e. first grasses and then woody types, is reduced, before that of pasture and bare soil. This preference to maintain pasture and bare soil was implemented as pasture provides food for livestock and because it was assumed that

converting bare areas into cropland is the most work- and resource-intensive way to increase the share of cultivated area. The cover fractions of cultivated areas were initialised with values corresponding to the year 2005. The extent of cultivated areas, irrigated and non irrigated, was obtained from data provided by the Land-Use Harmonization project (Hurtt et al., 2011), and the division into irrigated and rainfed crops was done according to the Global Map of Irrigation Areas (Fig. S1) (Siebert et al., 2005, 2013).

### 4.1.2 Soil characteristics and soil constraints

To account for the soil quality of a given grid-box, the soil constraints of the IIASA/ FAO's Global Agroecological Zones (version 3.0 (IIASA and FAO, 2012), and version 1.0 (IIASA and FAO, 2000) for permafrost areas) were translated into the surface fraction unsuitable for farming (Fig. S1). We considered nutrient retention capacity, rooting conditions, oxygen availability to

roots, excess salt, toxicity and workability as well as terrain slope constraints. As the study's focus is on the climate imposed limitations, the nutrient content of the soil was not taken into account. Here, it can be assumed that sufficient fertilizers are available for the investigated period even with the simulated increases in cultivated area (Vance, 2001; Cordell et al., 2009; Elser and Bennett, 2011; Dawson and Hilton, 2011). However, for long-term sustainability it is vital to eliminate the loss of terrestrial phosphorus, e.g. optimizing the application of fertilizers, adapting farming methods such as no-tillage and recaptur-

ing and recycling phosphorus. The cautious handling of fertilizers is not only key with respect to the available resources, as phosphorus is, effectively, a non-renewable resource (on the global scale the replenishment of phosphorus in the soil occurs on geological time scales). It is also essential for minimizing agriculture's detrimental impacts on other ecosystems, where an

excess of the respective elements can have grave consequences (Rockström et al., 2009).

The soil constraints were used to determine the maximum grid box fraction suitable for farming but they did not have a direct impact on the distribution of the simulated soil characteristics. This is a major limitation of the model, as the part of the grid box that is able to support crops should be represented by more favourable soil characteristics than the fraction that is affected by soil constraints. However, as the MPI-ESM is a global model, in which the soil characteristics are described on the large scale, subgrid-scale variability can not be taken into account consistently. The effect of soil variability within a grid

box is only taken into account for the calculation of surface runoff and infiltration (Düllmenil and Tondini, 1992). The soil data used in the MPI-ESM are based on adjusted FAO soil type and soil profile datasets and an overview over the derivation of soil parameters can be found in Hagemann and Stacke (2015). With this parameter set, the MPI-ESM captures the land surface water and energy fluxes well (Hagemann et al., 2013) but it should be noted that many of the soil parameters are only poorly constrained and the impact of the respective uncertainty is not well understood (Orth et al., 2016). Here, studies have shown

that the the large uncertainty in global soil data does not only introduces substantial uncertainties with respect to numercial weather prediction and climate simulations, but also with respect to simulated crop yields (Grassini et al., 2015; Folberth et al., 2016; Hoffmann et al., 2016; Montzka et al., 2017).

Finally, it should be noted that the information on soil constraints merely provides an upper bound to the cultivable area

in the present idealized scenarios but not necessarily to potentials in the real world. Especially in marginal areas, the cost of cultivation, e.g. due to the required irrigation related infrastructure or fertilizer input, may mean that agricultural intensification is not feasible under socio-economic considerations, even though it may be technically possible.

### 4.1.3   Irrigation

In the irrigated tiles, water is applied at every 450-second time-step during which irrigation is active. An irrigation period starts whenever the grid-box's reservoir is full and there are still sufficient $°C$ days left, here assumed to be 1300 $°C$ days, before temperatures are too low to grow crops. The later condition is evaluated based on the previous years distribution of $°C$ days.

Irrigation is stopped whenever the reservoir is empty or when temperatures, at or below the surface, become to cold. In the scheme, irrigation is simulated by increasing the root-zone soil moisture to the level at which plants make optimal use of the available water, i.e. in JSBACH at 75 % of the field capacity. This representation of irrigation constitutes a strong simplification of the range of different irrigation techniques. However, the irrigation requirements of around 3000 $km^3 year^{-1}$ simulated for the present-day irrigation of the reference simulations correspond well with estimates of other studies(Yoshikawa et al., 2013).

### 4.1.4   Harvest

Crops are harvested whenever they have accumulated a certain amount of biomass. The threshold for the harvest was set at an accumulated NPP of 25 $mol(CO2)m^{-2}$(canopy) or 300 $g(C)m^{-2}$(canopy). Making very broad assumptions about the crops that

are being represented in the model, this can be converted to a dry yield of $250\,\mathrm{t\,km^{-2}}$(canopy) (Li et al., 2014). For the harvest, the leaf area was reset to a minimum fraction and the biomass was transferred from the plants' carbon pool to a specific harvest pool.

## 4.2 Model: Water management Scheme

Irrigation-based agriculture constitutes about 70% of humanity's fresh water demand, while municipal, industrial and other agricultural demands constitute about 30% (Wada et al., 2013). Here, irrigation and non-irrigation water withdrawals differ in one important aspect. While irrigation is mostly a consumptive water use, i.e. water is transpired by plants or evaporated from bare soil areas, the largest fraction of the water used by other sectors is returned to the river after its use (Falkenmark and Lannerstad, 2005). Thus, when assuming that all waste water is treated to the extent that it can be reused, non-irrigation requirements only have a minor effect on the overall volume of utilizable water and the water management scheme represents them only indirectly (see below).

In the present version of the MPI-ESM, lakes do not exist on the subgrid-scale and for the standard resolution of T63, only a few very large lakes are accounted for in the model. Additionally, aquifers are not explicitly represented, and subsurface flows, together with the surface runoff, constitute the lateral inflow into the river network. Hence, the entire precipitation that reaches the terrestrial surface and is neither stored in the soil's pore space nor given back to the atmosphere, is eventually routed towards the oceans via the streamflow. This makes modelling the water withdrawals in JSBACH straightforward, as the entire accessible freshwater is combined within the discharge of rivers.

### 4.2.1 Environmental flow requirements

Water for irrigation can be withdrawn from a river, whenever the streamflow surpasses the environmental flow requirements. These are assumed to amount to 30% of the long-term mean flow (30-year-mean) (Smakhtin et al., 2004; Gerten et al., 2013; Pastor et al., 2014), or rather the long-term streamflow that would have occurred if water hadn't been withdrawn. Note that, because the consumptive water use of other sectors is not explicitly represented, the fraction of water that may be withdrawn for irrigation was reduced to 65% instead of 70%, to not overestimate the water availability. To approximate the streamflow that would have occurred without water withdrawals, the amount of water that enters the channel in each grid-box is accumulated along the river, i.e. from the source regions to the estuary. Aggregated over a longer period, here a year, this gives a good estimate of the volume of water that entered the river in all upstream grid-boxes and that, without withdrawals, would have been discharged through the river within this given grid-box.

### 4.2.2 Withdrawal and Storage

The fraction of the streamflow that exceeds the environmental flow requirements can be withdrawn and stored within a reservoir. The size of the reservoirs is determined such that it can hold the utilizable flow of a two month period. The respective estimates

are based on the streamflow and the environmental flow requirements of the previous years. Additionally, the reservoir size

is capped at $1.0\,\mathrm{m^3 m^{-2}}$(GBA), and for those years in which close downstream neighbours can not ensure the environmental flow, the reservoir size can not be increased but rather is decreased by $1\,\%$. Conceptually, the reservoir is evenly divided into an operational space and a buffer volume, each with a maximum size of $0.5\,\mathrm{m^3 m^{-2}}$(GBA). The size of the irrigated area and the decision to start irrigation are determined not based on the actual reservoir size but on the size of the operational space, and the buffer space merely functions to cushion the peak discharge that can occur as a result of extreme precipitation events.

Note that the MPI-ESM does not represent open water evaporation from water bodies on the subgrid scale, such as rivers, lakes or reservoirs. This results in a slight overestimation of the irrigation water availability and a slight underestimation of the terrestrial evapotranspiration. On global average around 3% of the water stored in reservoirs is evaporated annually, but this value can increase to as much as 10 % in arid regions (Falkenmark and Lannerstad, 2005).

### 4.3  Simulations

In all simulations, the MPI-ESM was run using a temporal resolution of 450 seconds, a vertical resolution of 47 atmospheric model levels, 40 vertical levels in the ocean and 5 vertical soil layers in all terrestrial grid-boxes. The horizontal resolution was T63 ($1.9° \times 1.9°$), which corresponds to a grid-spacing of about $200\,\mathrm{km}$ in tropical latitudes. On the land surface, spatial sub-grid scale heterogeneity was represented by 15 tiles, which were aggregated using a simple flux-aggregation scheme (Polcher et al., 1998; Best et al., 2004; de Vrese and Hagemann, 2016). Besides the 11 tiles that represent natural vegetation,

pasture and bare soil areas, rainfed C3 crops, rainfed C4 crops, irrigated C3 crops and irrigated C4 crops are each represented by an individual tile. The model's dynamical vegetation scheme was active in all the simulations and in the non-cultivated part of the grid-box, the cover fractions representing natural vegetation were calculated based on the plants's productivity and their natural and disturbance-driven mortality (Brovkin et al., 2009) .

In total 14 simulations were performed covering the period 1995 - 2114. These correspond to 5 sets of two simulations in which the two simulations that constitute a set differ only due to slight alterations in the initial conditions. The two simulations

were compared to evaluate the simulated impacts against the model's internal variability. In the analysis we always show the mean of the two simulations and for simplicity refer to this mean as one simulation. We simulated 3 sets, i.e. IR26, IR45 and IR85, in which the crop and water management schemes are both active and that differ only due to the prescribed GHG concentrations, i.e. RCP2.6, RCP4.5 and RCP8.5. To be able to quantify the impact of irrigation, we performed one set of simulations that were forced according to RCP4.5 in which only the crop management scheme was active but with the irrigation

scheme disabled, i.e. RF45. To quantify impacts due to climate change without CO2 fertilization, we conducted 4 additional simulations (IR26*, IR45*,IR85* and RF45*) that were identical to IR26, IR45,IR85 and RF45, but with the plant available CO2 limited to 380.0 ppmv. Comparing the above simulations to the present-day conditions of a reference set (REF) allows to estimate the potential change in cultivated area, food supply and climate for the respective scenarios.

In these reference simulations, the distribution of irrigated and non-irrigated crops was prescribed based on their present-day extent (Fig. S1)(Hurtt et al., 2011; Siebert et al., 2005, 2013). Note that the respective data correspond to the year 2005 and all comparisons to "present-day" or "current" conditions are made with respect to this year and not 2017. Furthermore, it was assumed that the water supply is not limited to the fraction of the streamflow which exceeds environmental flow requirements. Whenever the irrigation demands surpass the available water in these simulations, the deficit is provided by adding water to the

system. This approach is not conserving the water balance, but it is consistent with the assumption that, at present, irrigation requirements are partly satisfied from non-sustainable sources which are not represented in the model.

## 4.4 Analysis

### 4.4.1 Translation of cropland productivity into the sustainable population size

As mentioned above, in JSBACH crops are not represented by individual species such as maize, wheat or soy, but by two

functional types (C3 and C4 crops). This leads to an oversimplification of the biophysical response of crops and presents a strong limitation of the model in comparison to current day crop models. However, it is the common practice in Earth-System modelling. As a consequence, there is no detailed competition between different species and the ratio between NPP and crop yield is constant over time. Consequently, the relative yield increase is directly proportional to the relative increase in NPP. Furthermore, the preferential treatment of pasture resulted in the NPP of these types to remain largely at or ,when accounting

for CO2 fertilization, above present-day rates. Thus, it was assumed that the amount of calories produced by growing cattle on these lands remained constant in the simulations. As fish stocks can not be modelled with the MPI-ESM it was assumed that also the amount of food provided by this sector remains constant over time. Thus, by assuming that in 2005 80 % of our caloric intake was directly or indirectly supplied by crops (Steinfeld et al., 2006; Nellemann, 2009; FAO, 2016a), the global food supply relative to the year 2005 can be estimated as a function of the crops' simulated productivity.

To estimate the sustainable population ($K_{hum}$) based on the relative food supply, it was assumed that the relative increase in $K_{hum}$ is directly proportional to the relative increase in food production, i.e. that the ratio of food supply and population

5  remains at the level of the year 2005. Thus, $K_{hum}$ is not a measure of the planet's carrying capacity but merely the population size that can be sustained by the potential food supply when assuming present-day dietary patterns and food losses. It should be noted that it is not clear how the ratio of food production and population may develop in the future. For example, until 2050, dietary shifts are expected to increase the per capita demand for crops by more than 50% (Tilman et al., 2011), however it is highly uncertain how dietary patterns may develop once food becomes a limiting factor on the global scale. Additionally, it is

10  also uncertain to which extent we are capable of reducing food losses and waste, which at present make up almost a third of the calories produced (Nellemann, 2009; Godfray et al., 2010; Gustavsson et al., 2011; Lipinski et al., 2013; FAO, IFAD and WFP, 2015).

We also did not account for potential changes in the ratio of food to energy crops. Here, studies indicate that an increased demand for biofuels could result in a larger fraction of agricultural areas being dedicated to growing energy crops in the future (Berndes et al., 2003; SIMS et al., 2006; Johansson and Azar, 2007; Rathmann et al., 2010; Harvey and Pilgrim, 2011). As a first order effect, it can be assumed that the decrease in the yield of food crops is proportional to the increase in the share of energy crops. However, it is very unlikely that the same crops will be used to produce food and energy, as is the predominant practice at the moment. On the long term, it is more likely that dedicated plants, especially C4 grasses, would be used to produce energy (Heaton et al., 2008). Increasing the share of these plants would have an effect on climate that is different to the expansion of traditional (mostly C3) crops. Consequently, capturing the full effect of an increased demand for energy crops, requires their explicit representation in the model and further assumptions about the respective future demand.

### 4.4.2   Protected areas and the most productive 15% of the land surface

We did not perform dedicated simulations in which the expansion of cropland was limited to 15% of the ice-free land surface. Instead, we selected the grid-boxes that exhibited the highest productivity relative to the crops' canopy area, which disregards potential remote feed-back effects. We also did not simulate protected areas directly and land was cultivated to the extent that soil and climatic conditions allow it. The analysis with respect to protected areas was done by applying a fractional mask to the model output. For this mask we combined the areas that are at present under protection (Juffe-Bignoli et al., 2014; UN, 2016), i.e. as of 2016, and all areas that are covered by tropical forests at the beginning of the simulation (Fig. S1). To make the analysis more restrictive it was also assumed that the conversion of barren lands is too resource intensive. Consequently, the cover fraction of the bare soil tile, as of 2005, was included in the mask. Applying this fractional mask to the simulated cover fractions of the crop tiles allowed to estimate the ratio by which the cultivated area surpasses the habitable, non-protected fraction of a grid-box. This ratio was then used to scale the model output for the analysis. Note that there is no consensus which areas should be protected and, for the present study, it was assumed that at minimum areas placed under protection (as of 2016) (UN, 2016) as well as those covered by tropical forests (as of 2005) should be treated as protected areas.

*Code and data availability.*  Upon, possible publication the primary data will be made available via the German Climate Computing Center's long-term archive for documentation data. The model, scripts used in the analysis and other supplementary information that may be useful in reproducing the authors' work are archived by the Max Planck Institute for Meteorology and can be obtained by contacting publications@mpimet.mpg.de.

*Author contributions.*  P.d.V designed experiment, performed model adaptation, conducted simulations and analysis, T.S. performed model adaptation and was involved in the analysis and S.H. was involved in experiment design and conducting simulations. All authors reviewed the manuscript.

10   *Competing interests.*  The authors declare that they have no competing financial interest.

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

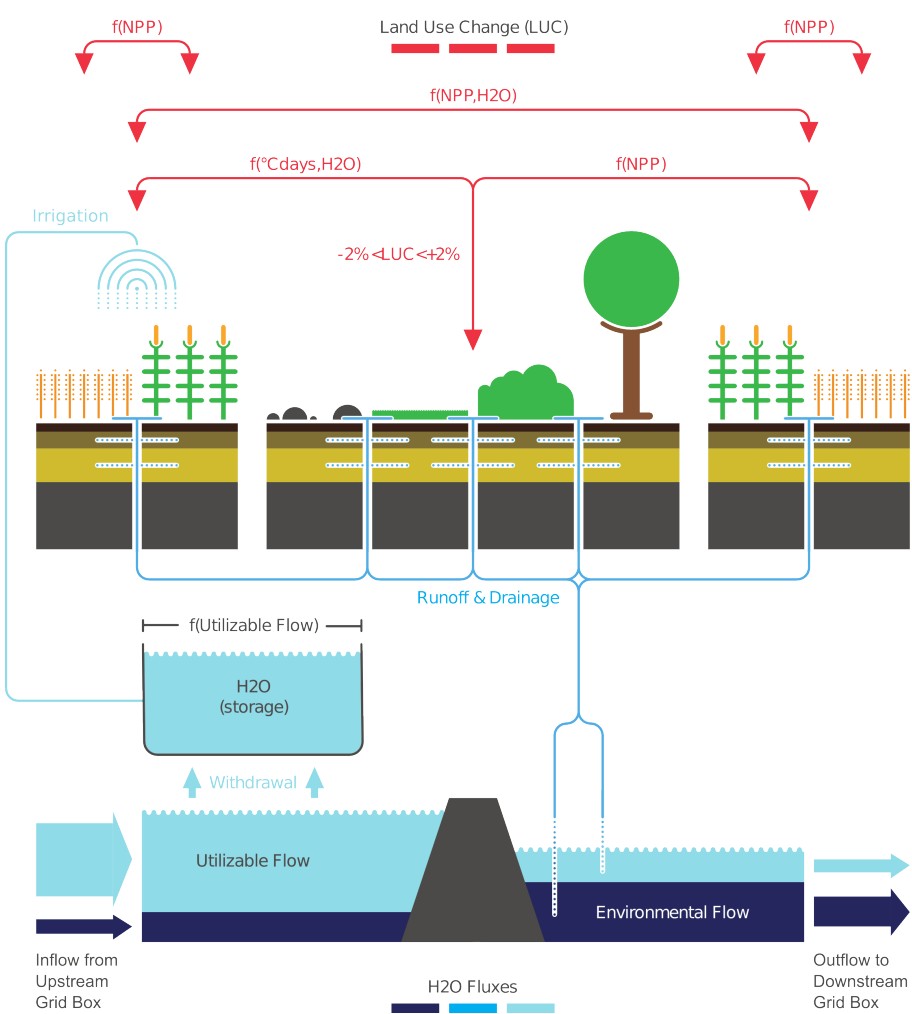

**Figure 1. Newly implemented processes**

**Cultivation scheme**

Within a given grid box and year, up to 2.2 % of the area can be converted from natural vegetation, pasture and bare soil to cultivated area. The rate at which the extent of croplands in- or decreases is modelled as a function of the crops' net primary productivity (NPP), in case of rainfed crops, and of the available water (H2O) and the length of the growing season (°C days), for irrigated crops. Areas can also be transferred from irrigated to rainfed crops and vice versa. This transfer is modelled based on the plants' NPP and the available water, in order to facilitate the expansion of the more productive technique (rainfed or irrigated) and to minimize the loss of agricultural output arising from decreasing irrigation water supplies. In a given grid box either C3 or C4 crops are being grown, depending on which of the types has the higher productivity under present climate conditions, measured by previous year's NPP.

**Water management scheme** On the terrestrial surface, all water which is not stored in the soil's pore spaces or evaporated and transpired forms runoff and drainage. These constitute the lateral inflow into the river within a given grid-box, from where water is accessible for agricultural use. About a third of the runoff and drainage that enters the river may not be withdrawn within any of the downstream grid boxes as it is required to sustain the ecological stability of the river, i.e. the environmental flow. The flow that surpasses these requirements, i.e. the utilizable flow, can be withdrawn, stored within a reservoir and be used for irrigation. The size of this reservoir (H20) is determined in such a way that it can store the utilizable flow of a two month period. **26**

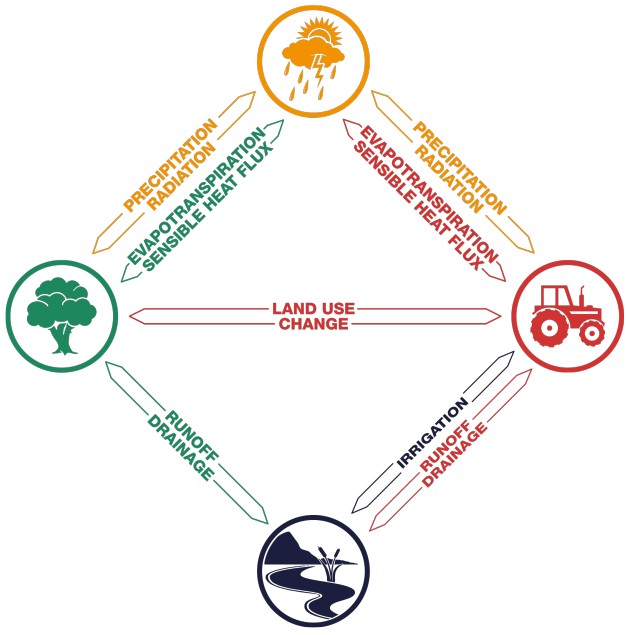

**Figure 2. Agriculture-climate-interactions**

The expansion of cultivated areas, i.e. land-use change, alters the characteristics of the terrestrial surface. It modifies the albedo, soil and vegetation characteristics which determine the exchange of energy, water and momentum between the land surface and the atmosphere. The consequent changes in the atmosphere's energy and moisture content as well as shifts in wind patterns influence cloud formation, hence the distribution of precipitation and radiation. The latter are key factors in the terrestrial hydrological cycle and determine evaporation, runoff formation and infiltration of water at the surface. Consequently they strongly affect the amount of water that is available for plants in form of soil moisture and also the volume of water stored in terrestrial water bodies, the latter of which determines the irrigation water supply. The availability of energy, i.e. solar radiation, and water, i.e. precipitation and irrigation, in turn, determines the productivity of plants, defining the regions which sustain farming and where suitable lands can be converted into croplands.

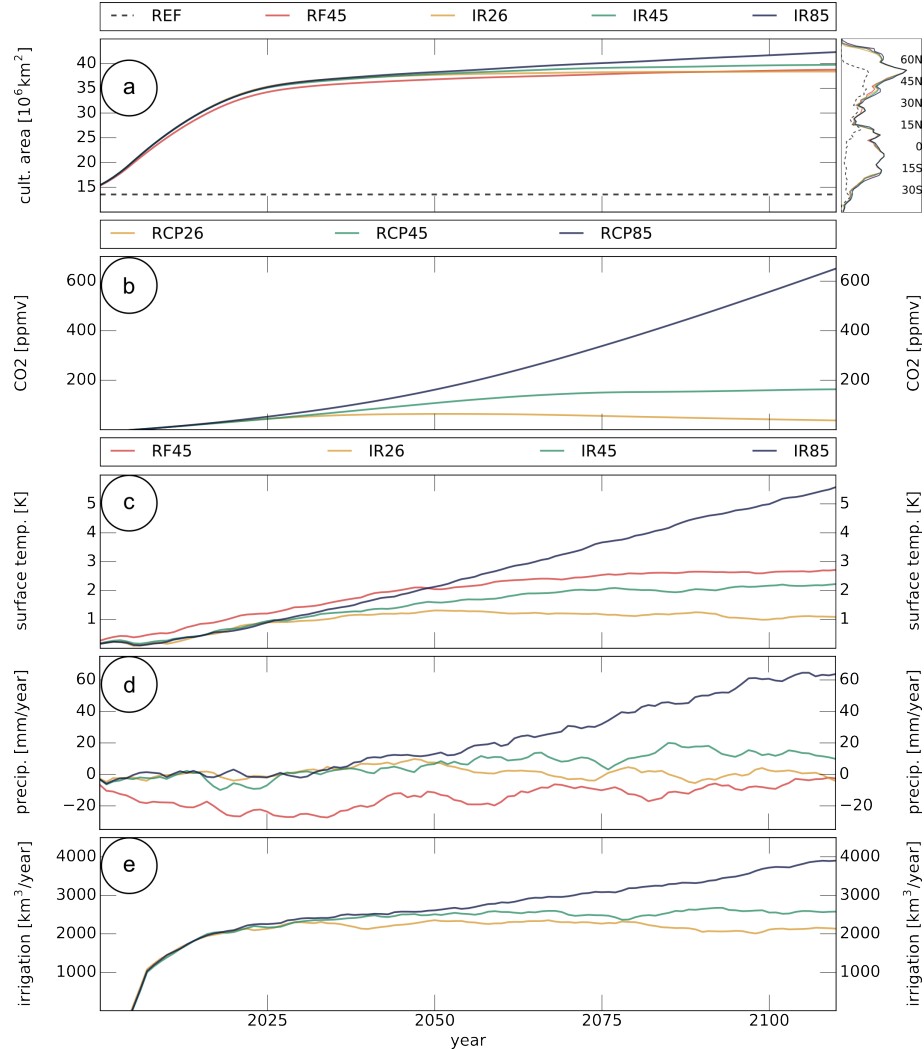

**Figure 3. Cultivated area and Global mean climate**

**a)** 10-year running mean of the global extent of cultivated areas. The panel on the right shows the zonal distribution of cultivated areas in 2100 (2085-2114 mean). **b)** Shows the increase in atmospheric $CO_2$ for the three GHG scenarios. For the land surface, **c**, **d** and **e** show the differences to present-day climate (REF); **c)** temperature, **d)** precipitation and **e)** accumulated irrigation.

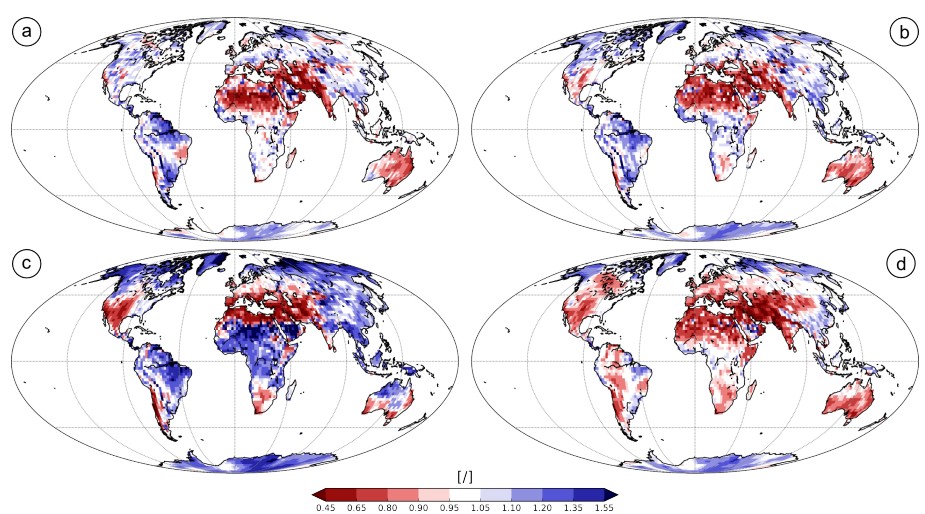

**Figure 4. Changes in water availability**

For IR26, **a)** shows the water available for crops (precipitation + irrigation) in 2100 relative to present-day (REF). **b)** Same as *a* but for IR45. **c)** same as *a* but for IR85. **d)** same as *a* but for RF45. Future water availability is reduced in those catchments in which precipitation declines, e.g. in the US, the Middle East and southern Europe, but it can also be reduced despite increases in precipitation. In India for example, irrigation is reduced substantially when rates are adjusted to a sustainable level and for IR26 (*a*) and IR45 (*b*) there is a predominant decrease in water availability despite increases in monsoon precipitation. For IR85 the precipitation increase is so strong that the in 2100 the water availability exceeds that of the year 2005 (*c*). Note that at long rivers, the available water near the estuary can also be reduced by increasing withdrawals by upstream riparian states, e.g. at the Nile, as present-day water right agreements are not represented in the model. The increase in water availability occurs mainly in South America and Sub-Saharan Africa, while between 15°North and 45°North amounts predominantly decline. Without irrigation, the water availability is reduced globally, with a few exceptions mostly in high northern latitudes and South East Asia (*d*).

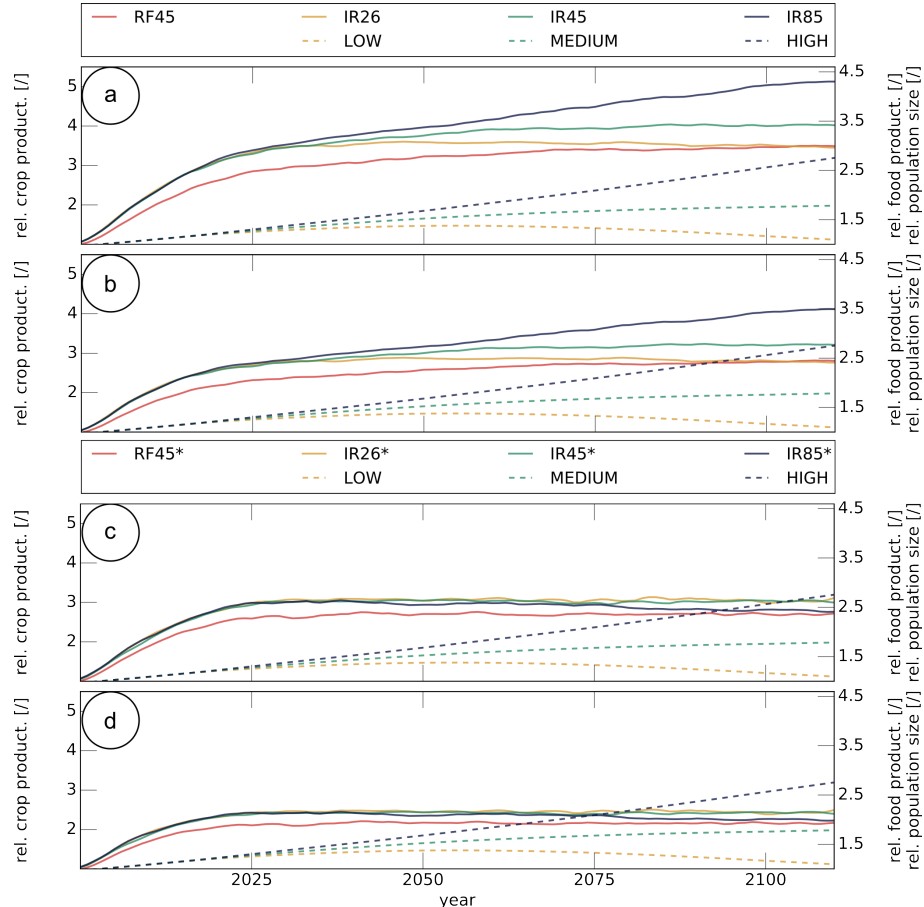

**Figure 5. Global food supply and UN population development scenarios**

(10-year running mean:) **a)** Left axis: solid lines indicate the global aggregate production of croplands relative to present-day productivity (2000-2009 mean; REF). Right axis: solid lines indicate relative global food production. Dashed lines (right axis) indicate the population size of the UN fertility scenarios (UN, 2015a) relative to the global population in 2005. **b)** Same as *a* but excluding protected areas.**c)** Same as *a* but without the CFE.**d)** Same as *b* but without the CFE.

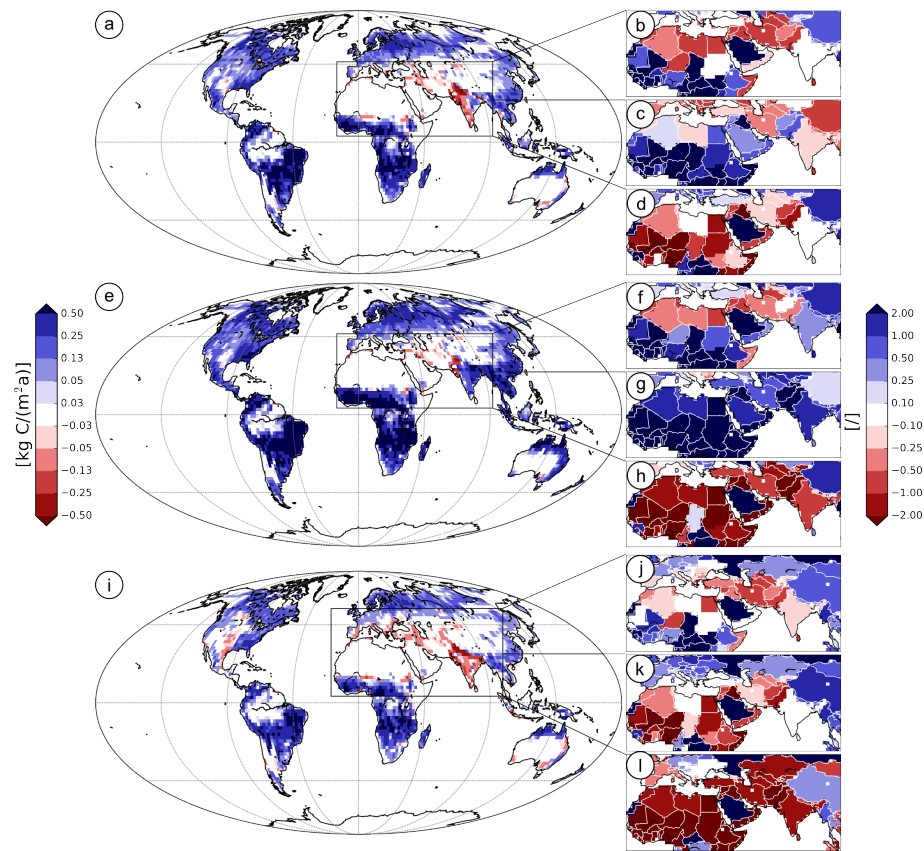

**Figure 6. Agricultural land, regional food production and population**

**a)** Productivity differences between the year 2100 in the RCP2.6 simulation and present-day (REF). The changes in net primary productivity are aggregated to the total (grid-box) area, so they show the combined effect of changes in the productivity of crops and changes in the spatial extent of croplands. **b)** Relative difference in food production between 2100 (IR26) and present-day (REF). **c)** Relative population change until 2100 for the high-fertility scenario (UN, 2015a). **d)** Difference between the relative changes in food supply and in population (note that to account for the share of imports in the overall supply, we assumed constant import levels (corresponding to the year 2005), and the food supply for net-importers was estimated from the domestic food production and the cereal import dependency ratio (FAO, 2016b)). **e,f,g,h)** Same as *a,b,c* and *d* but for RCP8.5 and high-fertility scenario. **i,j)** Same as *e* and *f* but without the CFE effect. **k)** Difference between the relative changes in food supply and in population for the low-fertility and low life expectancy scenario. **l)** Same as *k* but for the high-fertility and high life expectancy scenario.

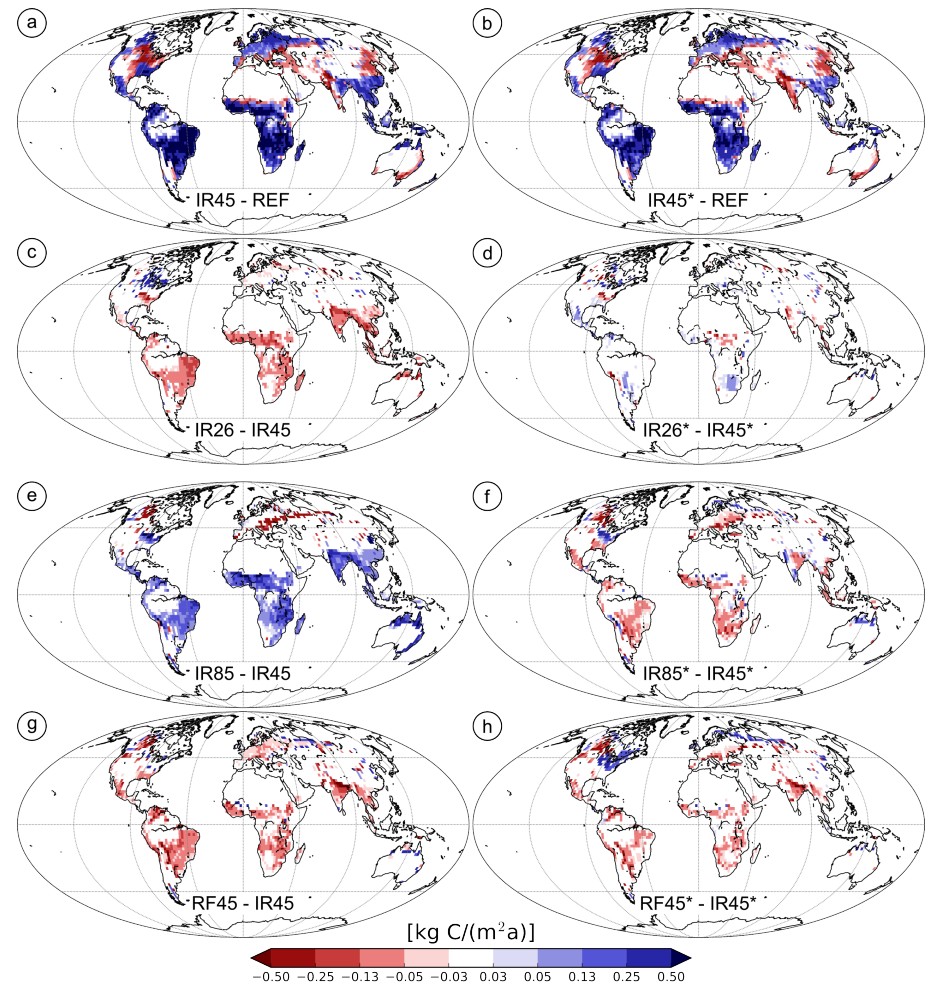

**Figure 7. Productivity in the most productive 15% of the ice-free land surface**

**a)** Shows the difference between the net primary productivity in 2100 (IR45) and present-day (REF), when the extent of future croplands is limited to the most productive 15% of the ice-free land surface. For IR45 the global productivity of croplands increases by a factor of 2.5 ($NPP_{Rel,2005}^{IR45}$=2.5). **b)** Same as *a* but without the CFE ($NPP_{Rel,2005}^{IR45*}$=1.9). **c)** Shows the differences in productivity between the RCP2.6 and the RCP4.5 scenario ($NPP_{Rel,2005}^{IR26}$=2.2). **d)** Same as *c* but without the CFE ($NPP_{Rel,2005}^{IR26*}$=2.0). **e)** Same as *c* but for IR85 and IR45 ($NPP_{Rel,2005}^{IR85}$=3.1). **f)** Same as *e* but without the CFE ($NPP_{Rel,2005}^{IR85*}$=1.8). **g)** Same as *c* but for RF45 and IR45 ($NPP_{Rel,2005}^{RF45}$=2.2). **h)** Same as *g* but without the CFE ($NPP_{Rel,2005}^{RF45*}$=1.7)