# Peer review of "Exploring the biogeophysical limits of global food production under different climate change scenarios"

_Earth System Dynamics, 2017_

## Referee Comment (RC1) · Anonymous Referee #1 · 19 Dec 2017

This paper provides an interesting analysis of the maximum global extent of cropland area and maximum crop yields on such areas under conditions of future climate change, increasing $CO_2$ concentration, and some consideration of ecosystem conservation. Given that an advanced earth system model with an incorporated land surface scheme was applied, the analysis appears to be solid from the point of view of biophysical process representations. However I have some reservations about the framing, the manuscript structure, and the interpretation of some results. They are detailed in the following and should be critically reflected in a revision of the manuscript.

A main point of concern with the current manuscript is that it provides little information on the methods and scenario assumptions, unless one reads the Methods section. The Introduction ends with a short summary but then the Results immediately follow –

more information is required at this point because it is crucial for readers' understanding and interpretation of the findings, and to make transparent the (partly strong) scenario assumptions. I suggest to at least add the following information at this place: what is the spatial/temporal resolution and the forcing of the model (incl. climate, CO2, land use/protected areas); what are the environmental flow requirements about; how is area converted into (future) food supply; how is it possible that areas decline, what is the criterion for that. Furthermore, a clear research question should be formulated.

Related to that, parts of the Results section should be formulated more carefully. For example, on page 3 it is stated that "almost three quarters of all cultivable land could be farmed by the beginning of the next century". Also on page 7 line 16: is a "vast increase in food imports" really the only way out, can such a claim be supported by other literature? Please make always sure that this is only in your very idealized simulation, which explores some upper potentials based on biophysical processes and land-climate feedbacks but not on socially (and technologically?) feasible potentials.

The Discussion is very short (with the two first paragraphs being only an extension of the Results) and rather weak. Here I would expect a critical reflection of scenario assumptions including more literature references on 1) how does the CO2 effect in your model, which has such a very strong impact on the results (increasing K_hum by up to 12 billion!), relate to findings from other studies; 2) what are the crop management assumptions in your study, which is important relative to other studies which use specific increases in management which in turn affects the crop area

Minor / technical comments:

The title is quite general and does not well reflect that it is about a global modelling study of theoretical maximum potentials, thus I suggest to adapt it in this regard.

Abstract: Some more crucial information should be added, that is, which (climate) scenario runs you analyzed, why areas are to be abandoned in some of the simulations, and what are the "most optimistic assumptions" mentioned in the final words.

The first paragraph of the results is partly Methods, partly self-evident, it could be deleted.

Does the CO2 fertilisation effect apply to both crops and natural vegetation?

Page 4 first paragraph: What do you mean, "without requiring any previous changes"?

Same paragraph and at other places: I think the term "sustainable" is not correctly used here, it is misleading; rather use "achievable"?

Page 4 third paragraph: It is unclear whether you here talk about global sums only or about regional patterns (i.e. is increased demand met globally or in the very regions); in any case more focus and examples on specific regions are needed.

Same page, next paragraph: I do not understand why "the results highlight the importance...".

Page 5 first paragraph: 20% or 0.5K temperature increase is high – especially if that is a global value? What particular scenario does this relate to, i.e. how large an area is assumed to be irrigated and where? I also think it is not correct to express temperature changes in %.

Page 6 line 10: Please avoid such a statement if possible, as it appears to "recommend" RCP8.5 because it increases food production; this is also in contrast to many climate change impact studies that suggest strong declines in yields – which need to be cited here (or rather in an extended Discussion).

Same page lines 21-23: I do not understand this, safe climatic range for food...?
* * *

---

## Referee Comment (RC2) · Anonymous Referee #2 · 25 Jan 2018

General comments:

Manuscript deals with future projection of both irrigated and rain fed cropland expansion and food production as responding to changing climate and availability of land and water resources. Existing Earth System model (ESM) was used in the study and adapted for the analysis of bio-physical feedbacks between climate, land cover (use), and water resources at the global scale. Study provides estimates of future (2100) cropland expansion (both under irrigated and rain fed conditions), future crop yield and corresponding sustainable population size with three representative GHG concentration pathways (RCP2.6, RCP4.5, and RCP8.2) also assuming for $CO_2$ fertilization effect as an important source of uncertainty in global yield estimates. With this scope and complexity of the system studied the paper is well within the scope of the ESD journal.

[Figure]

Two new modules were introduced into existing ESM for cropland expansion dynamics including irrigation and calculation of available water resources used for irrigation. This is obviously novel approach to dynamic land cover and land use change modelling as a direct element of ESM yet same time introducing new sources of uncertainties of simulated climate change and biophysical response of the agricultural systems. Study concludes that under climate change, the cultivated land could be nearly tripled and can sustain population of 15 to 27 bn people (compared to current 6.5 bn) which brings very important and sound message for wide community of scientists and practitioners and as such it should be carefully discussed so that it does not lead to any inaccurate interpretations (see also specific comments). Study uses standard and well established methods and all assumptions on newly introduced modules are well outlined and referred to existing published data or knowledge. ESM simulation results which reflect a combination of RCP and crop management and CO2 concentration scenarios provide enough results to come up with all conclusions and interpretations presented in the paper. The study provides general conceptual overview of the methods and algorithms used to come up with results and the input data and assumptions are described in enough detail to understand the work the authors did. The primary data and the ESM and simulation outputs post-processing codes are available to interested parts on request. The work authors did is put well in the context and provides enough credit to other and preceding works. In my opinion the title of the paper is not well chosen because speaking about limits of food production the study should address more comprehensive view also including other than biophysical constrains. More straightforward title reflecting the main highlight of the study which is cropland expansion and sustainable world population would better reflect the content of the study. Abstract brings good and concise overview of the paper bringing all important methodological assumptions and the most important findings of the study. The paper is written fluently and reads well.

Specific comments/questions:
Comparison of selected climate variables (CO2 concentration, surface temperature, precipitation, and water deficit) to original ESM would nicely emphasize the importance of the newly introduced crop management and water management modules.

How can be competition with crops produced for energy taken into account?

The biophysical assumptions on CO2 fertilization effect are only valid if sufficient amount of nutrients is supplied to the crops which is not case in most developing parts of the world

Yet making sense from the bio-physical perspective the projected cropland expansion (or loss) should be also examined in the socio-economic development concept

Possible gains of cropland areas in marginal areas could be not suitable for intensifications and/or not accessible or effective from the socio-economic or geopolitical point of view. Not mentioning this explicitly could lead to wrong message and too optimistic estimates of global food production and carrying capacity for growing worldwide population in the future. In this sense the SSPs should be at least briefly discussed in the context of the presented study not just compared to existing official population estimates released by UN.

Food production is simulated mostly as a function of water availability for the plants driven mostly by the climate, but locally affected by soil water holding capacity and soil water balance. There is no information on the source of soil data used in simulations nor the discussions on possible effects of soil variability on the crop yield production (c.f. e.g. Folberth et al. 2016, NatCom)

Minor limitation of the approach is also oversimplification of biophysical response of the crops which can be better addressed by other, more specific models – a comparison of the simulated potential yields with other global gridded crop models would make the modelling outcomes more reliable.

---

## Author Response (AR1)

**Climate change imposed limitations on potential food production: Reply to reviewers and editor**

Philipp de Vrese[1], Tobias Stacke[1], and Stefan Hagemann[2]

[1]Max Planck Institute for Meteorology, The Land in the Earth System, Hamburg, 20146, Germany
[2]Helmholtz-Zentrum Geesthacht, Institute of Coastal Research, Geesthacht, 21502, Germany

*Correspondence to:* Philipp de Vrese (philipp.de-vrese@mpimet.mpg.de)

We structured our reply to the reviewer's comments as follows. At first we repeat the **referees point of criticism** in bold letters, which is followed by a reply which is not included in the manuscript (in plain letters), and finally we give the *parts in the manuscript that were altered* in italics.

5 **Reply to reviewer 1**

**1.1) A main point of concern with the current manuscript is that it provides little information on the methods and scenario assumptions, unless one reads the Methods section. The Introduction ends with a short summary but then the Results immediately follow - more information is required at this point because it is crucial for readers' understanding and interpretation of the findings, and to make transparent the (partly strong) scenario assumptions. I suggest to at**
10 **least add the following information at this place: what is the spatial/temporal resolution and the forcing of the model (incl. climate, CO2, land use/protected areas); what are the environmental flow requirements about; how is area converted into (future) food supply; how is it possible that areas decline, what is the criterion for that.**

Due to the description of the new schemes, the methods section became rather long and we were hoping to avoid the more
15 traditional structure in which it follows the introductory section. We are grateful to the reviewer for pointing out the missing details, which allows us to provide the reader with a good overview without having to read the entire methods section. To include the additional information, we changed the introduction to the following:

*In the approach, the spatial extent of cultivated areas is modelled as a function of climatic conditions as well as the agricul-*
20 *tural water supply. In regions where conditions allow for at least a minimum productivity, i.e. the crops' net primary productivity (NPP) corresponds to a yield of at least $\approx 250\,t\,km^{-2}(canopy)\,year^{-1}$, the cultivated area is extended incrementally until all cultivable areas are occupied, i.e. the land not limited by soil or terrain constraints. In regions in which the NPP falls below this threshold, the area under crops declines. The NPP was also used to estimate the potential food production, by assuming that the changes in crop yields are proportional to changes in the plants' NPP. To estimate the potential food production on a*

*hydrologically sustainable basis, future water withdrawals are limited to the fraction of renewable fresh water which exceeds environmental requirements. Here, it is assumed that about a third of the long-term mean flow is required to ensure ecological stability and may not be withdrawn (Pastor et al., 2014). Water for irrigation is removed from the river network and stored in a dedicated reservoir. When required, the water is applied to the soil, from where it evaporates, is taken up by plants and*
5 *transpired or returned to the river via subsurface runoff (for more details on the methodology see Sec. 4). Together with the changes in the surface-atmosphere exchange of energy and moisture that result from alterations of the surface characteristics, this closes the feedback loop between land-use and climate (Fig. 2).*

*We used this adapted model to investigate the climate-agriculture dynamics during the 21st century that result from the*
10 *maximization of the cropland area under different atmospheric green house gas (GHG) concentration scenarios (Fig. 3b, Tab. 1 and Sec. 4). The simulations cover the period 1995 - 2114 and were forced according to three representative concentration pathways (RCP, Meinshausen et al. 2011; van Vuuren et al. 2011) that assume a peak and a subsequent decline of emissions until 2020 (RCP2.6) and 2040 (RCP4.5) as well as an ongoing increase in emissions (RCP8.5). They use a temporal resolution of 450 seconds, a horizontal resolution of T63 (1.9° × 1.9°) and vertical resolution of 47 atmospheric model levels.*

To clarify which areas are considered to be under protection we added the following:

*By excluding these areas from the analysis, i.e. areas placed under protection (Juffe-Bignoli et al., 2014; UN, 2016) and those covered by tropical forests (as of 2005), the cultivated area in 2100 is reduced by roughly 15% (Sec. 4 and Fig. S2).*

With respect to the "climate forcing" we did not alter the manuscript, as we performed fully coupled simulations (land, ocean, atmosphere) and, besides the green house gas concentrations, the only external forcing is given by the prescribed orbital parameters.

25 **1.2) Furthermore, a clear research question should be formulated.**

Even though we did not formulate it as a question, we tried to present the target of the investigation more concisely:

*The focus of this investigation is on the global crop yields that are achievable under future climate conditions and, in the*
30 *following analysis, we will show the potential expansion of cultivated areas, the changes in global yields and how these relate to future food security. The effects of changes in irrigated and rainfed cropland area on climate will only be discussed very briefly as their detailed analysis goes beyond the scope of this study.*

**1.3) Related to that, parts of the Results section should be formulated more carefully. For example, on page 3 it is**
35 **stated that "almost three quarters of all cultivable land could be farmed by the beginning of the next century". Also on**

**page 7 line 16: is a "vast increase in food imports" really the only way out, can such a claim be supported by other literature? Please make always sure that this is only in your very idealized simulation, which explores some upper potentials based on biophysical processes and land-climate feedbacks but not on socially (and technologically?) feasible potentials.**

We edited the sentence on page 3 and in the revised version it starts by stating that the cropland expansion pertains to the simulations.

*In the simulations, the cropland area can be tripled to roughly 38 - 42 $\cdot 10^6 km^2$, and almost three quarters of all cultivable land can be farmed by the beginning of the next century (Fig. 3a, Tab. 1)*

It is true that the study uses some simplifying assumptions which could turn out to be overly pessimistic. However, we also neglected certain constraints which could become decisive limitations in the real world. To maintain a balance between admitting that the study may underestimated crop yields (i.e. a "vast increase in food imports" is not required) and not presenting an overly optimistic perspective, we included the following part on page 7.

*It is possible that the present study underestimates the potential food production especially as possible technical solutions, such as better adapted crops or large scale desalination efforts, are not being accounted for. On the other hand, the study neglects important constraints, e.g. resulting from fertilizer availability or the limited water-use efficiency of irrigation systems. As, in reality, these will strongly affect future crop yields, the present idealized scenario may likely provide an overly optimistic outlook. This is especially the case for the simulations that assume a large increase in GHG concentrations, i.e. RCP4.5 and RCP8.5, and account for the full benefits of the CFE.*

It wasn't our intention to present the results as pertaining to real-world potentials. In order to make sure that they are understood as merely idealized scenarios, we included the following part at the end of the introductory section.

*It should be noted that the present framework targets biogeophysical feedbacks, with a special focus on the hydrological cycle, while other important limitations arising due to social, political, economic and technological factors are being neglected. Therefore, the below results merely pertain to the development of cropland areas and yields in an idealized scenario and not necessarily to real-world potentials, the latter of which may be much more constrained by e.g fertilizer availability, cost of transport and irrigation related infrastructure, dietary shifts and the competition with energy crops.*

**1.4 a) The Discussion is very short (with the two first paragraphs being only an extension of the Results) and rather weak. Here I would expect a critical reflection of scenario assumptions including more literature references on 1) how does the CO2 effect in your model, which has such a very strong impact on the results (increasing K_hum by up to 12**

**billion!), relate to findings from other studies; ...**

We agree that the CFE should have been discussed in more detail. We hope to correct this by including the below discussion:

5    *Another reason for the high simulated yields, is the model's comparativly strong CFE. Many studies have investigated the effect of increasing atmospheric $CO_2$ concentrations on vegetation (Tubiello et al., 2007). These indicate that there is a substantial photosynthetic response to increasing $CO_2$ levels, i.e. under optimal conditions, doubling the present day $CO_2$ concentrations leads to an increase in photosynthesis of 30 % - 50 % for C3 and 10 % - 25 % for C4 plants. With respect to crop yields, the existing studies exhibit large uncertainties and strong variations between crop types and regions. For $CO_2$*

10   *increases similar to the ones assumed by the RCP4.5 scenario, the estimates range from a 2.5 % to a 25 % yield increase per 100 ppmv increase in $CO_2$ (Amthor, 2001; Tubiello et al., 2007; Ainsworth et al., 2008; Asseng et al., 2013; McGrath and Lobell, 2013). In the RCP8.5 scenario, the atmospheric $CO_2$ concentrations towards the end of the century exceed 1000 ppmv. At these levels, the benefits due to additional $CO_2$ are much smaller as even C3 crops are close to (or have already reached) their saturation level. For the rise in $CO_2$ concentrations assumed by this scenario the average yield increase is expected to*

15   *be below 6 % - 8 % per 100 ppmv increase in $CO_2$ (Parry, 1990; Amthor, 2001; Ainsworth and Rogers, 2007; Ainsworth and McGrath, 2010). In comparison to these studies, which predominantly consider yield increases under optimal conditions, the MPI-ESM simulates a very strong CFE (approximated by the productivity difference between the simulations with and those without increasing the plant-available $CO_2$). In regions that are being farmed at present (grid boxes in which 5 % of the area or more were covered by crops in the year 2005), the simulations for the RCP4.5 scenario that account for irrigation exhibit*

20   *an average increase in yield per area of about 18% per 100 ppmv increase in $CO_2$. Owing to the higher temperatures and lower water availability, the simulated strength of the CFE is slightly lower, i.e. about 14% per 100 ppmv increase in $CO_2$, when irrigation is not represented. For the RCP8.5 scenario, our simulations showed an increase of about 10 % per 100 ppmv increase in $CO_2$. These values place the CFE simulated with the MPI-ESM at the higher end of the range of current estimates, in case of the RCP8.5 scenario even exceeding them, indicating that the model overestimates the strength of the CFE and the*

25   *resulting crop yields.*

**1.4 b) ... 2) what are the crop management assumptions in your study, which is important relative to other studies which use specific increases in management which in turn affects the crop area.**

30   Possibly the most important assumption is that future irrigation is being maximized within sustainable limits. On one hand this directly increases the water availability, on the other hand it leads to climatic conditions that are much more favourable for plants, i.e. lower temperatures and increased precipitation. When removing these effects, i.e. focusing on the simulation without irrigation and on regions that are dominated by rainfed agriculture, our results actually agree well with other studies. In the manuscript, we included the following part into the discussion section:

*In the study, the crop's general response to changes in climate agrees well with estimates of other studies (Lobell et al., 2011; Asseng et al., 2014; Challinor et al., 2014). When omitting the CFE and effects of irrigation (RF45\*), regions that are presently dominated by rainfed agriculture exhibit an average decline in yield per area of about 5 % per K temperature increase, i.e. in grid boxes where 5 % of the area or more were covered by crops in the year 2005 and less than a third of this cropland area was irrigated, a temperature rise of about 2.6 K caused an average reduction in crop yields per area of about 12 %. The yield response to changes in temperature is strongly affected by the study's management assumptions and, in the same regions, the average yield per area increases by about 2 % per K temperature increase, when irrigation is maximized within sustainable limits (IR45\*), i.e. for the temperature rise of about 2.1 K we estimated an average increase in crop yields per area of about 5 %. Hence, the assumptions made with respect to future irrigation, including the representation of the resulting climate feedbacks, are one of the reasons why the development of global crop yields under the RCP scenarios is much more positive than in many other studies (Guoju et al., 2005).*

**1.5) The title is quite general and does not well reflect that it is about a global modelling study of theoretical maximum potentials, thus I suggest to adapt it in this regard.**

We changed the title to the following (see also reviewer 2; point 2.1):

*"Exploring the biogeophysical limits of global food production under different climate change scenarios"*

**1.6 a) Abstract: Some more crucial information should be added, that is, which (climate) scenario runs you analyzed,**

We extended the abstract to include the following information:

*For three green house gas concentration scenarios (RCP2.6,RCP4.5,RCP8.5), we show that the total cropland area could be extended substantially throughout the 21st century, especially in South America and sub-Saharan Africa, where the rising water demand resulting from increasing temperatures can largely be met by increasing precipitation and irrigation rates.*

**1.6 b) ... why areas are to be abandoned in some of the simulations, ...**

*When accounting for the $CO_2$ fertilization effect, only few agricultural areas have to be abandoned owing to declines in productivity, while increasing temperatures allow to expand croplands even into high northern latitudes.*

**1.6 c) ... and what are the "most optimistic assumptions" mentioned in the final words.**

Admittedly, "optimistic" may have been a poor choice of words. We changed the formulation to the following:

*For certain regions the situation is even more concerning and guaranteeing food security in dry areas in Northern Africa, the Middle East and South Asia will become increasingly difficult, even for the idealized scenarios investigated in this study.*

**1.7) The first paragraph of the results is partly Methods, partly self-evident, it could be deleted.**

With restructuring the introductory section to include the additional details, this part has been removed from the results section.

**1.8) Does the CO2 fertilisation effect apply to both crops and natural vegetation?**

The CFE applies to managed as well as to natural vegetation and by limiting the plant-available $CO_2$ to 380 ppmv also the natural vegetation is affected.

**1.9 a) Page 4 first paragraph: What do you mean, "without requiring any previous changes"?**

We ment to say that the climatic conditions, i.e. temperature and precipitation, are already suitable for growing crops at the beginning of the simulation. In order to clarify this, we edited the sentence to the following:

*Wide areas could be cultivated without requiring any changes in the conditions, i.e. temperatures and precipitation rates are already in a favourable range at the beginning of the century, and the largest potential for expansion is given in latitudinal zones in which crops are already being grown (Fig. 3a; right panel).*

**1.9 b) Same paragraph and at other places: I think the term "sustainable" is not correctly used here, it is misleading; rather use "achievable"?**

We are not entirely sure that "achievable" correctly describes the simulated irrigation withdrawals, as it gives the impression that really all available water is being used. We designed the water management scheme in a way that the environmental flow is being ensured. Thus, from an ecological perspective the simulated irrigation is sustainable, and we did not change the manuscript.

**1.10 a) Page 4 third paragraph: It is unclear whether you here talk about global sums only or about regional patterns (i.e. is increased demand met globally or in the very regions); in any case more focus and examples on specific regions**

**are needed.**

Indeed, it was bit unclear that we were talking about the general behaviour on the land surface. To clarify this, we edited this paragraph (see below). For the RCP4.5 and RCP8.5 it is valid to omit a more detailed regional analysis as it is really only
5   very few grid boxes whose behaviour deviates from the description. For RCP2.6, however, there are a few areas in which the increased water demand can not be met, which we added to the manuscript.

*The scenarios that exhibit a strong temperature rise also show a substantial increase in precipitation over land (Fig. 3d). For RCP4.5 (IR45) mean precipitation rates increase by up to 20 mm year$^{-1}$ and in IR85 they increase by about 60 mm year$^{-1}$,*
10   *which amounts to more than 8% of the terrestrial precipitation as of 2005. Increased precipitation rates do not only reduce the water stress for rainfed crops, but between 2025 and 2100 they also increase the water available for irrigation; globally by roughly 500 km$^3$ year$^{-1}$, for IR45, and by almost 2000 km$^3$ year$^{-1}$ for IR85 (Fig. 3e). As a consequence, the increased water demand of irrigated and rainfed crops resulting from higher temperatures can be met to the extent that, after 2025, there are only very few areas in the world in which farming becomes unsustainable. This however is only the case when fully accounting*
15   *for the potential benefits due to the CO$_2$ fertilization effect (CFE; see below). For the simulations with only a small increase in GHG concentrations (IR26) there is no permanent increase in precipitation, i.e. after a peak in the 2040s the rates decline to their initial levels, while the average temperature at the land surface increases by $\approx$1K. Here, the plant's increasing water requirements can not be met everywhere and in some dry regions in South and Central Asia, the Sahel zone and Australia farming becomes unsustainable after 2025 and cropland areas have to be abandoned.*

**1.10 b) Same page, next paragraph: I do not understand why "the results highlight the importance...".**

Again, this may not have been an ideal choice of words. We changed the paragraph to:

25   *The results show that future climate is substantially impacted by the maximization of irrigation within sustainable limits.*

**1.11 a) Page 5 first paragraph: 20% or 0.5K temperature increase is high ? especially if that is a global value? What particular scenario does this relate to, i.e. how large an area is assumed to be irrigated and where?**

30   We estimated the impact of irrigation as the difference between the RCP4.5 simulations with (IR45) and without (RF45) irrigation. With about 0.5 K, the temperature effect due to irrigation is indeed very large, but so is the irrigated area. In the IR45 simulation the irrigated area is more than quadrupled from 2% to about 8% of the global land surface (as compared to the reference simulation). Here, Fig. S4 in the supplementary material gives a good overview of how the irrigated area develops when it is being maximized within sustainable limits (even though it pertains to the year 2025 and irrigation still increases
35   afterwards). In the text we clarified that the reduced temperature refers to the effects of irrigation with respect to the RCP4.5

scenario:

*Furthermore, for the RCP4.5 scenario, irrigation reduces the simulated 21st-century temperature increase by almost 20% ($\approx 0.5$ K averaged over the global land surface; Fig. 3c), and in irrigated regions the effect can amount to several K.*

**1.11 b) I also think it is not correct to express temperature changes in %.**

Due to the arbitrary zero point of the common temperature scales (F & C) changes in temperature should indeed not be given in %, e.g. global surface temperature changes by 5%. However, in the manuscript we do not have the zero point issue as we merely give the fraction by which a certain temperature increase is reduced, which, to the best of our knowledge, is a valid formulation.

**1.12 a) Page 6 line 10: Please avoid such a statement if possible, as it appears to "recommend" RCP8.5 because it increases food production; ...**

We made the respective formulation more careful as we did not want to give the impression of promoting the RCP8.5 scenario.

*Here, the results seem to suggest that the high concentration trajectory is favourable with respect to food production, however, this is only the case if the CFE is as efficient as simulated by the MPI-ESM.*

**1.12 b) ... this is also in contrast to many climate change impact studies that suggest strong declines in yields - which need to be cited here (or rather in an extended Discussion).**

It is quite difficult to compare our results to other models as they are substantially impacted by our ability to include climate feedbacks and the maximization of irrigation. The only valid comparison that can be made is for yields in the simulation without irrigation (RF45*) and that only in regions that, at present, are dominated by rainfed crops. In these regions, the simulated yield decline of about 5%/K is actually in good agreement with other studies which we included in the discussion section (see also point 1.4 b).

*In the study, the crop's general response to changes in climate agrees well with estimates of other studies (Lobell et al., 2011; Asseng et al., 2014; Challinor et al., 2014). When omitting the CFE and effects of irrigation (RF45*), regions that are presently dominated by rainfed agriculture exhibit an average decline in yield per area of about 5 % per K temperature increase, i.e. in grid boxes where 5 % of the area or more were covered by crops in the year 2005 and less than a third of this cropland area was irrigated, a temperature rise of about 2.6 K caused an average reduction in crop yields per area of about 12 %. The yield*

*response to changes in temperature is strongly affected by the study's management assumptions and, in the same regions, the average yield per area increases by about 2 % per K temperature increase, when irrigation is maximized within sustainable limits (IR45*), i.e. for the temperature rise of about 2.1 K we estimated an average increase in crop yields per area of about 5 %. Hence, the assumptions made with respect to future irrigation, including the representation of the resulting climate feedbacks, are one of the reasons why the development of global crop yields under the RCP scenarios is much more positive than in many other studies (Guoju et al., 2005).*

**1.12 c) Same page lines 21-23: I do not understand this, safe climatic range for food ... ?**

To clarify, we changed the formulation to:

*Given the high level of uncertainty connected to the CFE, the range of climatic conditions that are favourable for food production is likely limited to the conditions resulting from the RCP4.5 scenario.*

**Reply to reviewer 2**

**2.1) In my opinion the title of the paper is not well chosen because speaking about limits of food production the study should address more comprehensive view also including other than biophysical constrains. More straightforward title reflecting the main highlight of the study which is cropland expansion and sustainable world population would better reflect the content of the study.**

To make the title more precise, especially with respect to being an investigation of biophysical mechanisms, we changed it to the following:

*"Exploring the biogeophysical limits of global food production under different climate change scenarios"*

**2.2) Comparison of selected climate variables (CO2 concentration, surface temperature, precipitation, and water deficit) to original ESM would nicely emphasize the importance of the newly introduced crop management and water management modules.**

We fully agree with the reviewer that a further analysis of the climate impacts of the simulated cropland expansion (especially irrigated) would add additional insights to the manuscript. However, we think that the comparison to standard MPI-ESM simulations should not stand by itself but requires an extensive discussion, which is beyond the scope of this paper. This is especially the case as the present setup is not only new in the MPI-ESM but has also not been investigated by any other modelling group (an extreme scenario that is interactively constrained by water and land availability). There is an ongoing study

that targets the possible importance of irrigation as a geo-engineering tool, and we plan to comprehensibly discuss the climate impacts in the present simulations as a part of this study.

**2.3) How can be competition with crops produced for energy taken into account?**

Unfortunately, the impact of this competition can not be fully evaluated using the present model setup. As herbaceous biomass plantations would have a very distinct effect on climate, we would have to adapt the model to also represent them explicitly. In the method section (Translation of cropland productivity into the sustainable population size) we explained that the assumption of a constant ratio between food and energy crops constitutes a possible oversimplification.

*We also did not account for potential changes in the ratio of food to energy crops. Here, studies indicate that an increased demand for biofuels could result in a larger fraction of agricultural areas being dedicated to growing energy crops in the future (Berndes et al., 2003; SIMS et al., 2006; Johansson and Azar, 2007; Rathmann et al., 2010; Harvey and Pilgrim, 2011). As a first order effect, it can be assumed that the decrease in the yield of food crops is proportional to the increase in the share of energy crops. However, it is very unlikely that the same crops will be used to produce food and energy, as is the predominant practice at the moment. On the long term, it is more likely that dedicated plants, especially C4 grasses, would be used to produce energy (Heaton et al., 2008). Increasing the share of these plants would have an effect on climate that is different to the expansion of traditional (mostly C3) crops. Consequently, capturing the full effect of an increased demand for energy crops, requires their explicit representation in the model and further assumptions about the respective future demand.*

**2.4) The biophysical assumptions on CO2 fertilization effect are only valid if sufficient amount of nutrients is supplied to the crops which is not case in most developing parts of the world. Yet making sense from the bio-physical perspective the projected cropland expansion (or loss) should be also examined in the socio-economic development concept.**

It is true that fertilizer availability is an issue especially for developing countries but we fear that going into a detailed discussion on this problem on a regional or even national level goes beyond the scope of this study. Nonetheless, we fully agree with the reviewer that nutrient limitations are a key factor which we may not have acknowledged sufficiently in our discussion of the results. In order to discuss the nutrient supply as a global scale issue, we integrated the following aspects into the discussion section.

*In addition to climate effects, weed and insect pests as well as increasing nutrient requirements are expected to reduce the strength of the CFE (Tubiello et al., 2007). Here, constraints due to fertilizer availability present one of the key limiting factors. For example, Rosenzweig et al. (2014) investigated the crop yield response for the RCP8.5 scenario as simulated with different global gridded crop models. The study showed that yields for the major crop types predominantly increase if no explicit nitrogen limitation was accounted for. However, when nitrogen limitations are introduced and fertilizer application is restricted to*

*present day rates, the effect of CO2 fertilisation is greatly reduced and all major types exhibit a decline in crop yields through-*
*out the low and parts of the mid latitudes (Rosenzweig et al., 2014). In principle di-nitrogen gas provides an unlimited source*
*of nitrogen. However, nitrogen fixation, i.e. the process by which atmospheric nitrogen is made available for plants, requires*
*high energy inputs. At present, the share of fertilizer production in the global energy consumption is estimated to be around*
5 *1% (Vance, 2001; Dawson and Hilton, 2011), and the fertilizer requirements as proposed in this study could easily increase*
*this share to more than 5%. In case of phosphorus the situation is more difficult as it is, effectively, a non-renewable resource*
*and our supply stems from mines which are located in only a few countries. The size of the phosphate rock deposits is highly*
*uncertain and by far the largest deposits have only since recent been included when taking stock. Given our current use of*
*phosphorus, these known resources would last for the next 400 - 800 years (Cordell et al., 2009; Dawson and Hilton, 2011).*
10 *With the increase in fertilizer demand, as proposed by this study, the deposits of phosphate rock may not last long beyond*
*the investigated period. Industrial agriculture, even on the present-day scale, is not possible without phosphorus fertilization*
*and productivity would quickly diminish to the level prior to the agricultural revolutions of the 19th and 20th century if our*
*resources are exhausted. Hence, the future food supply will strongly depend on how much energy is available for the production*
*of fertilizers and how effectively nutrients can be recycled.*

*But even if sufficient fertilizers could be provided, this would increase other problems related to their application. The*
*present-day fertilizer use already has strong detrimental impacts on the ecosystems in certain regions, where an excess of the*
*respective elements can leave entire lakes, rivers and coastal stretches uninhabitable to plants and animals (Vitousek et al.,*
*1997; Smith, 2003; Rockström et al., 2009). As a consequence, it has been suggested that the extent of croplands should not*
20 *surpass 15% of the global ice-free land surface (Rockström et al., 2009; Steffen et al., 2015). Any further expansion could*
*bring the planet to a tipping point, e.g due to hypertrophication resulting from increased use of fertilizers and the loss of*
*biodiversity. This would mean that we have to retain agricultural expansion far below the limits set by climatic conditions.*
*Limiting croplands to 15% of the global ice-free land surface, would roughly halve the potential cropland area as estimated*
*by this study, i.e. about a third of the ice-free land surface, resulting in similar decreases in crop yields and food security.*
25 *Additionally, the study's assumption that per capita food requirements will remain at present day levels may also contribute to*
*an overestimation of the level of food security. Dietary shifts are expected to double global food requirements by 2050 while*
*the population is only expected to increase to about 9bn (Godfray et al., 2010). It is highly doubtful whether this dietary shift*
*and population increase could be sustained without expanding the cultivated areas beyond the safe limit of 15%. Here, our*
*results indicate that only a very strong CFE could lead to the necessary increase in crop yields. Additionally, it would require*
30 *shifting cultivated areas to the most productive regions, mostly in sub-Saharan Africa, South America and South East Asia,*
*and to provide an almost perfect irrigation system (Fig. 7).*

**2.5 a) Possible gains of cropland areas in marginal areas could be not suitable for intensifications and/or not accessi-**
**ble or effective from the socio-economic or geopolitical point of view. Not mentioning this explicitly could lead to wrong**
35 **message and too optimistic estimates of global food production and carrying capacity for growing worldwide population**

**in the future.**

To not run the risk of presenting an overly optimistic outlook, we added the following passage to the description of the soil constraints:

*Finally, it should be noted that the information on soil constraints merely provides an upper bound to the cultivable area in the present idealized scenarios but not necessarily to potentials in the real world. Especially in marginal areas, the cost of cultivation, e.g. due to the required irrigation related infrastructure or fertilizer input, may mean that agricultural intensification is not feasible under socio-economic considerations, even though it may be technically possible.*

**2.5 b) ... In this sense the SSPs should be at least briefly discussed in the context of the presented study not just compared to existing official population estimates released by UN.**

To provide some more information on the RCPs, we describe the assumptions made with respect to population development, future energy demand and the mix of energy carriers. Furthermore, we contrast the underlying land-use change scenarios with our own findings.

*The RCPs are consistent with distinct socio-economic pathways that differ strongly with respect to future energy demand and the mix of energy carriers. Included are assumptions about resource availability and climate policies, which determine the contribution of fossil fuels to the energy mix, as well as assumptions about the population development, which strongly affects future energy demands. Additionally, the scenarios take into account different land-cover and land-use change projections which reflect future food and energy demands as well as policies with respect to reforestation (Meinshausen et al., 2011; van Vuuren et al., 2011). RCP2.6 and RCP4.5 present intermediate scenarios with ambitious emission reductions, which in case of RCP2.6 even include a decline in the use of oil. For RCP2.6 and RCP4.5 the population development corresponds to UN projections assuming a low to medium fertility and life expectancy in the future (UN, 2004, 2015a, b). In contrast, RCP8.5 presents a highly energy-intensive scenario without the implementation of any climate policies. The high energy demand in this scenario partly results from a strong population growth, which corresponds to a medium to high population trajectory in the UN development scenarios. In order to estimate the level of food security for a given combination of RCP and population development scenario, the simulated $K_{hum}$ can be compared to the population levels proposed by the UN scenarios. Here, the simulations indicate that the ability to sustain future populations depends heavily on the strength of the CFE. When assuming the full benefits, only the population of the high-fertility (and life-expectancy) scenario may become unsustainable, i.e the respective population trend surpasses $K_{hum}$ as simulated for RCP2.6 and RCP4.5, and that only if protected areas are maintained (Fig. 5a,b). However, without the CFE the food requirements resulting from the high-fertility scenario can not be met by any simulated supply, even if protected areas are converted into croplands (Fig. 5c,d). Also the population of the medium-fertility scenario is very close to $K_{hum}$, indicating that we may need to cultivate almost all non-protected areas and*

*to have a near-perfect system for irrigation in order to meet the future food requirements of this scenario. Here, our findings contradict the RCP's underlying scenarios. These assume that the population increase in RCP8.5 could be sustained without increasing the cropland area beyond $20 \cdot 10^6 \, km^2$, while the population increase in RCP4.5 could even be met with a substantial decline in the cultivated area (Meinshausen et al., 2011; van Vuuren et al., 2011).*

**2.6) Food production is simulated mostly as a function of water availability for the plants driven mostly by the climate, but locally affected by soil water holding capacity and soil water balance. There is no information on the source of soil data used in simulations nor the discussions on possible effects of soil variability on the crop yield production (c.f. e.g. Folberth et al. 2016, NatCom).**

The uncertainty in soil parameters is an issue that is not only relevant with respect to crop yields but also for climate simulations, and at present we are participating in the Soil Parameter Model Intercomparison Project (SP-MIP), led by Lukas Gudmundsson and Mathias Cuntz, to better understand to which extent the large spread among LSMs with respect to water-balance variables is related to soil model parameters. For simulated crop yields the problem is potentially larger because also subgrid scale variability becomes an important factor as, on the subgrid-scale, there should be a correlation between presence of crops and favourable soil characteristics. We tried to point out the respective shortcomings by including the following passage into the description of the soil constraints:

*The soil constraints were used to determine the maximum grid box fraction suitable for farming but they did not have a direct impact on the distribution of the simulated soil characteristics. This is a major limitation of the model, as the part of the grid box that is able to support crops should be represented by more favourable soil characteristics than the fraction that is affected by soil constraints. However, as the MPI-ESM is a global model, in which the soil characteristics are described on the large scale, subgrid-scale variability can not be taken into account consistently. The effect of soil variability within a grid box is only taken into account for the calculation of surface runoff and infiltration (Düllmenil and Tondini, 1992). The soil data used in the MPI-ESM are based on adjusted FAO soil type and soil profile datasets and an overview over the derivation of soil parameters can be found in Hagemann and Stacke (2015). With this parameter set, the MPI-ESM captures the land surface water and energy fluxes well (Hagemann et al., 2013) but it should be noted that many of the soil parameters are only poorly constrained and the impact of the respective uncertainty is not well understood (Orth et al., 2016). Here, studies have shown that the the large uncertainty in global soil data does not only introduces substantial uncertainties with respect to numercial weather prediction and climate simulations, but also with respect to simulated crop yields (Grassini et al., 2015; Folberth et al., 2016; Hoffmann et al., 2016; Montzka et al., 2017).*

**2.7) Minor limitation of the approach is also oversimplification of biophysical response of the crops which can be better addressed by other, more specific models ? a comparison of the simulated potential yields with other global gridded**

**crop models would make the modelling outcomes more reliable.**

It is quite difficult to compare our results to other models as they are substantially impacted by our ability to include climate feedbacks and the maximization of irrigation. The only valid comparison that can be made is for yields in the simulation without irrigation (RF45*) and that only in regions that, at present, are dominated by rainfed crops. In these regions, the simulated yield decline of about 5%/K is actually in good agreement with other studies which we included in the discussion section (see also reviewer 1; point 1.4 b).

*In the study, the crop's general response to changes in climate agrees well with estimates of other studies (Lobell et al., 2011; Asseng et al., 2014; Challinor et al., 2014). When omitting the CFE and effects of irrigation (RF45*), regions that are presently dominated by rainfed agriculture exhibit an average decline in yield per area of about 5 % per K temperature increase, i.e. in grid boxes where 5 % of the area or more were covered by crops in the year 2005 and less than a third of this cropland area was irrigated, a temperature rise of about 2.6 K caused an average reduction in crop yields per area of about 12 %. The yield response to changes in temperature is strongly affected by the study's management assumptions and, in the same regions, the average yield per area increases by about 2 % per K temperature increase, when irrigation is maximized within sustainable limits (IR45*), i.e. for the temperature rise of about 2.1 K we estimated an average increase in crop yields per area of about 5 %. Hence, the assumptions made with respect to future irrigation, including the representation of the resulting climate feedbacks, are one of the reasons why the development of global crop yields under the RCP scenarios is much more positive than in many other studies (Guoju et al., 2005).*

Nonetheless, representing crops by just two types is a strong oversimplification and we changed the respective manuscript part to the following:

*As mentioned above, in JSBACH crops are not represented by individual species such as maize, wheat or soy, but by two functional types (C3 and C4 crops). This leads to an oversimplification of the biophysical response of crops and presents a strong limitation of the model in comparison to current day crop models. However, it is the common practice in Earth-System modelling.*

**Reply to editor**

Before answering to the comments, we would like to thank Prof. Lucht very much for his constructive criticism, especially as we understand that comments and advise by the editor can not be taken for granted.

**Despite improved wording that describes the nature of your study and guards against an overly optimistic interpretation of results, I believe you still treat a full 3-fold increase of the worlds agricultural area in too casual a manner. Agri-**

**culture is by far the main driver of environmental degradation, habitat loss and biodiversity decline today. Ecosphere stability is a critical issue because of these developments. Both at the poor end and at the industrial end unsustainable practices are wide-spread.The huge expansion of agricultural area you model under what you claim are criteria of sustainability is largely not driven by the dynamic climatic feedback effects your model is built to take into account but**

5    **by an enormous expansion of agriculture into land you deem available and suitable already under current climate (an expansion seen as being met essentially within the next decade, which already illustrates the highly artificial nature of the study). Technically, geophysically, there is a basis for that, as you argue, but this is from a mostly hydroclimatic view of the Earths ecosphere, not an actual consideration of sustainability or even functional ecosphere stability.**

10    There is no doubt that agriculture, even with its present day extent, has huge detrimental effects on the environment, and tripling the respective areas could easily push Earth beyond an ecological tipping point. However, it is very difficult to give an estimate for the acceptable extent of cultivated areas, e.g. as this also depends very much on which areas are protected and the land management (especially fertilizer use) of the cultivated areas. In the main part of the manuscript we decided to discuss the impact of preserving only the bare minimum extent of protected areas (tropical forests and areas that are already placed under

15    protection), as it best corresponds to the idealized scenario in which everything is subordinate to the goal of maximizing the cropland fraction. This however, does not mean that this expansion is necessarily in line with maintaining ecological stability. Therefore, we used the discussion section to show the impact of actually maintaining the extent of cultivated areas below 15% of the ice-free land surface which Rockström et al. (2009) give as a planetary boundary for land cover change. To clarify this we edited the respective part of the results section:

*This expansion would require converting areas into croplands regardless of their present function in the Earth system, including those whose cultivation is highly debatable with respect to biodiversity and terrestrial carbon stores, e.g. the Amazon rainforest. There is no consensus on the extent of protected areas required to maintain the planet's ecological stability and consequently also no consensus on the area that, from an ecosystem perspective, could be transformed into croplands. Merely*

25    *omitting areas presently placed under protection (Juffe-Bignoli et al., 2014; UN, 2016) and those covered by tropical forests from the analysis already reduces the cultivated area in 2100 by roughly 15% (Sec. 4 and Fig. S2). However, it is most likely that ensuring Earth's habitability will require the protection of a much larger fraction of the land surface (Sec. 3).*

**I think the limitations inherent to the approach suggest you should be explicit about the particular nature of the**

30    **study you conducted through your advance in modelling – I do not think it is the actual carrying capacity of Earth. Nonetheless, you readily translate this huge expansion of land under direct human use into a carrying capacity that is also several times larger than the current world population. Can you do that with any confidence? It is great progress that ESMs are beginning to enter this arena, however, the model approaches used are not superior to those used in the respective more disciplinary fields (that neglect the feedbacks) and therefore need to be treated with appropriate**

35    **caution. The strength of your approach, your core advantage, is a study of the feedbacks with the climate system. How-**

**ever, in agriculture these frequently are not the largest factor determining yields. What is more, in your manuscript you do not focus mostly on characterizing the effects of feedbacks, but equally on the overall total productivity, with implications e.g. for carrying capacity, which then requires confidence in the baseline results.**

We fully agree with the editor that our investigation does not account for all aspects that determine agricultural production, thus the study does not target Earth's actual carrying capacity as limited by the availability of food. However, to the best of our knowledge, it is not uncommon to focus on merely one aspect determining the carrying capacity which, in our case, is the carrying capacity as limited by climatic and hydrological limitations on food production. It is not our intention to hide the fact that we study just this one aspect, i.e. that we investigate highly idealized scenarios, and that a model-based estimate of Earth's actual carrying capacity would require the coupling of a broader range of dedicated models that incorporate a larger number of details in their respective fields (and even though the ESM may incorporate some of the processes of the specialized models, their representation is often superior in comparison to the ESM's, e.g. the representation of crops in the ESM vs. global crop models). We tried to make this even more prominent by changing the end of the introductory section to the following:

*The present framework targets biogeophysical feedbacks, with a special focus on the hydrological cycle, while other important social, political, economic, ecological and technological considerations are being neglected. Thus, it is important to note, that the below results merely pertain to the development of cropland areas and yields in a highly idealized scenario but not necessarily to real-world potentials, the latter of which may be much more limited by factors such as economic costs, fertilizer availability, the need to minimize environmental degradation, habitat and biodiversity loss, the limited capabilities of existing irrigation systems, dietary shifts and the competition between food and energy demands. An estimate of actual agricultural potentials requires a more comprehensive framework that is capable of representing all these factors, e.g. by integrating a broader range of models including dedicated crop, economic and ecosystem models.*

Additionally, we see how our use of the phrase sustainable might have been less than ideal. We intended to used it in the sense that a population can be supported with enough food. Of course, it additionally has the connotation of resources being used in an ecologically, economically and politically responsible way, especially as sustainability as a prominent management principle implies a holistic approach. In our study we can only claim that irrigation withdrawals are sustainable, in that they ensure the environmental flow requirements, and we edited the manuscript accordingly.

**I am not criticizing that you venture into these fields, this is great progress for ESMs, but I am as sceptical as perhaps the reviewers of the real-world veracity of the results obtained.**

It is true that, by themselves, the "carrying capacity estimates" of our study only have a limited real-world veracity, as they pertain to a highly idealized scenario. We hope that, with the edited introductory section this is now clearer for the reader. However, the results also demonstrate that there are strong feedbacks connected to the expansion of, especially irrigation based,

agricultural areas, many of which result in a more beneficial climate and help mitigate the detrimental effects of climate change (with respect to global crop yields, thus the carrying capacity). And, even though the absolute magnitude of these feedbacks is strongly depending on the actual expansion of the cropland area and irrigation, we do not see a reason for doubting that they will occur in reality. Furthermore, we show that even an optimized irrigation and a maximization of the cropland extent may not prevent the yields in certain densely populated areas from declining substantially. Given the (idealized) nature of the simulations these declines will likely be even more severe in reality.

**Despite your discussion of CFE and performance of runs without it, you heavily rely on it as a consequence you see the largest carrying capacity under business-as-usual climate warming. This does not match established findings of the detrimental consequences of high-end climate change and should lead to a more careful, critical discussions of your own results in your responses to the reviewers you tend to dismiss this aspect by stating that your results cannot really be compared to other studies and that you indeed see a decline in the no-CFE case.**

Here, we do not fully agree with the editor for the following reasons:

Not only in the discussion section, but also in the main text we tried to avoid indicating that the RCP.8.5 would be most beneficial for food production (despite the respective simulation producing the highest yield). Early on in the manuscript we stated that many of the (positive) aspects of the RCP8.5 simulation rely on the CFE for which "it is very uncertain which role it may play for future crop yields, and it is even possible that it's benefits will be balanced completely by other factors such as nitrogen limitations (Rosenzweig et al., 2014; Smith et al., 2015; Obermeier et al., 2016)". Therefore, we always additionally presented the results for simulations without the CFE and concluded that "Given the high level of uncertainty connected to the CFE, the range of climatic conditions that are favourable for food production is likely limited to the conditions resulting from the RCP4.5 scenario".

Furthermore, there are also more sophisticated crop/ vegetation models that show a predominant increase in productivity under the RCP8.5 scenario (see e.g. GAEZ-IMAGE, LPJ-GUESS, and LPJmL in Rosenzweig et al. (2014) Fig. 3 bottom right), even in mid and low latitudes. Here, it appears that one key factor determining whether RCP8.5 will lead to a de- or an increase in productivity is the availability of nitrogen. Therefore, we included a discussion not only on the potential supply of fertilizer but also on their detrimental impacts on the environment which ultimately led to the comparisons of future productivity if only 15% of the ice-free land surface are being used for farming.

Most importantly, we do not want to dismiss the reviewers' request for a comparison to other studies. We merely wanted to state that a comparison to other studies can not be done for all the simulations and also not for the entire land surface. A comparison only makes sense for the simulation that is not affected by an increase in irrigation and that only in those areas that are not strongly irrigated in the reference run. For this comparison however, the magnitude of the overall decrease in

productivity for the simulation without irrigation actually fits within the range of other estimates, leading us to believe that the crops biophysical reaction to increasing temperatures is actually represented well in the MPI-ESM and that the increase in (per area) yields in the irrigation simulations is in fact resulting from this process and the resulting feedbacks (see Fig. 1).

[Figure]

**Figure 1. Relative (per area) productivity in 2100 for RF45\* and IR45\***

**Left side:** Per area cropland productivity in the year 2100 as simulated for the RCP4.5 scenario without irrigation (RF45\*) relative to the productivity in the year 2005 (REF). **Right side:** Same as the left side, but for the RCP4.5 scenario when irrigation is included in the simulation (IR45\*); Grid boxes in which less than 5 % of the area or were covered by crops in the year 2005, or where more than than a third of the cropland area was irrigated are masked in grey.

[revised manuscript text omitted]

~~As demonstrated in this study, climate change and the need to limit water withdrawals to a sustainable levelwill affect our future food production, especially if the CFE should not have strong beneficial effects. While increasing temperatures provide the opportunity to expand croplands into the high northern latitudes , some densely populated areas may become heavily dependant upon food imports.At the same time, it is likely that we will 
[revised manuscript text omitted]